# Memory-Integrated Reconfigurable Adapters: A Unified Framework for Settings with Multiple Tasks

**Susmit Agrawal**[1,2,3*†]
susmit.agrawal@bethgelab.org

**Krishn Vishwas Kher**[1*]
cs19b23p000001@iith.ac.in

**Saksham Mittal**[1]
ai22btech11024@iith.ac.in

**Swarnim Maheshwari**[1]
cs25mtech02006@iith.ac.in

**Vineeth N. Balasubramanian**[1,4]
vineethnb@cse.iith.ac.in
vineeth.nb@microsoft.com

[1]IIT Hyderabad    [2]University of Tübingen    [3]Tübingen AI Center    [4]Microsoft Research, India

[*]Equal Contribution
[†]Majority of work done at IIT Hyderabad

## Abstract

Organisms constantly pivot between tasks such as evading predators, foraging, traversing rugged terrain, and socializing, often within milliseconds. Remarkably, they preserve knowledge of once-learned environments sans catastrophic forgetting, a phenomenon neuroscientists hypothesize, is due to a singular neural circuitry dynamically overlayed by neuromodulatory agents such as dopamine and acetylcholine. In parallel, deep learning research addresses analogous challenges via domain generalization (**DG**) and continual learning (**CL**), yet these methods remain siloed, despite the brain's ability to perform them seamlessly. In particular, prior work has not explored architectures involving associative memories (**AM**s), which are an integral part of biological systems, to jointly address these tasks. We propose Memory-Integrated Reconfigurable Adapters (**MIRA**), a unified framework that integrates Hopfield-style associative memory modules atop a shared backbone. These memory modules store adapter-weight updates as values and retrieve them via learned keys. Associative memory keys are learned post-hoc to index and retrieve an affine combination of stored adapter updates for any given task or domain on a per-sample basis. By varying only the task-specific objectives, we demonstrate that **MIRA** seamlessly accommodates domain shifts and sequential task exposures under one roof. Empirical evaluations on standard benchmarks confirm that our **AM**-augmented architecture significantly enhances adaptability and retention: in **DG**, **MIRA** achieves SoTA out-of-distribution accuracy, and in incremental learning settings, it outperforms architectures explicitly designed to handle catastrophic forgetting using generic **CL** algorithms. Extensive ablation studies validate the necessity of both associative memory storage and post-hoc key learning for robust interpolated retrieval of adapters. By unifying adapter-based modulation with biologically inspired associative memory, **MIRA** delivers rapid task switching and enduring knowledge retention in a single extensible architecture, charting a path toward more versatile and memory-augmented AI systems. [1]

---

[1]Project Page: https://snimm.github.io/mira_web/

39th Conference on Neural Information Processing Systems (NeurIPS 2025).

# 1   Introduction

Organisms across the animal kingdom navigate myriad environments and behavioral demands, flexibly switching between survival tasks (such as foraging for food or evading predators) and complex social interactions, within fractions of a second. Concrete examples include: echolocating bats, which adjust their sonar pulse rates from 20 to 200 Hz in milliseconds when tracking evasive prey, while simultaneously computing three-dimensional flight paths to avoid obstacles [25, 73], or jazz pianists among humans who instantaneously transition between playing a memorized composition and spontaneous improvisation, a cognitive shift marked by distinct prefrontal activation patterns [8, 61]. Similarly, many animals (including humans) learn to navigate a particular environment, such as intricate pathways of a dense forest, or subtle acoustic cues of a predator's approach, and retain that knowledge indefinitely, without the catastrophic forgetting that plagues AI systems [95, 21]. Such phenomena are commonly attributed to the brain's capability to rapidly repurpose the same circuitry for multiple tasks without dismantling its core wiring [18, 62, 9]. Some neuroscientific observations indicate the presence of overlapping sets of neurons that encode multiple task rules simultaneously [62, 71, 99], with neuromodulatory signals that regulate the active rule at a given time [51].

From another perspective, the field of deep learning has developed a rich taxonomy of paradigms that echo these natural behaviors. Domain generalization (DG) methods ensure robustness to distribution shifts [89]; for example, a driver-assistance model that has learned to drive during daytime adapts to driving at night. Domain-incremental learning (DIL) [38] seeks to learn and identify the same objects in new orientations, abstractions, and settings; for example, a model learns objects from cliparts and then learns to identify the same objects in anime or real world. Class-incremental learning (CIL) [38] aims to accumulate knowledge of newer classes as they arrive over time without forgetting; for example, a model that learns to identify flora and fauna being introduced to data from a new continent. Although these paradigms differ in terms of data availability, distributional shifts, and forgetting dynamics (and are treated so most commonly in literature), they share a common thread: adapting efficiently to new tasks or environments. Efforts in these paradigms have largely progressed in isolation, unlike in the brain where such adaptation tasks are handled conjointly [62, 71, 99]. We seek to address this gap in this work. An ancillary line of work on parameter-efficient fine-tuning (PEFT) has attempted adaptation with an objective of parameter efficiency by freezing a given base model and adapting it to new tasks or domains via small, task-specific "adapters". These adapters are overlaid over the base model to allow switching between different tasks. Techniques such as LoRA [36], VeRA [47], and FourierFT [23] instantiate this idea.

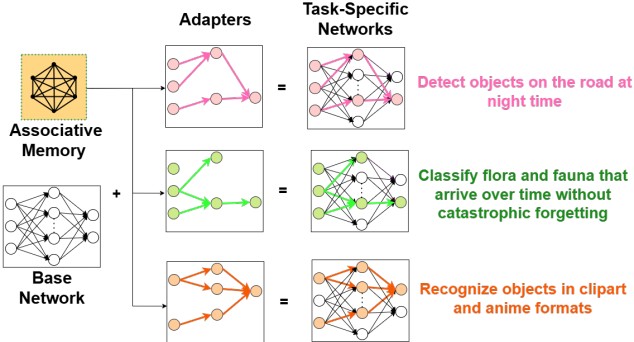

Figure 1: Associative memories can enable networks to quickly adapt to diverse tasks, by storing and recalling task-specific weights on-demand. **MIRA** proposes a framework for such an approach.

However, despite a conceptual resemblance to neural task-switching, existing work has predominantly overlooked explicit memory-based mechanisms that biology suggests are fundamental to rapid and efficient adaptation [44]. Motivated by this observation, we propose a novel architecture and learning methodology that explicitly integrates biologically plausible associative memory models into deep learning frameworks, as illustrated in Figure 1. Diverging from contemporary work in associative memories that stores raw data or their representations, our architecture stores weight adapters as values within an associative memory. Furthermore, instead of using fixed keys, we learn retrieval keys post hoc to optimally recall adapters for task-specific modulation of the substrate network. These keys facilitate accurate and context-sensitive retrieval through a Hopfield network, effectively generating affine combinations of adapter adjustments required for task-specific modulation at inference time, thus enhancing knowledge retention and out-of-domain generalization. Our proposed neuro-inspired framework thus establishes a common umbrella architecture capable of simultaneously and effectively addressing DG, CIL, and DIL scenarios. Our approach differentiates itself by serving all these settings through only minor adjustments in task-specific objective functions. Our key message in this work is to expound the utility of integration of associative memories into DL

architectures for improved efficacy on multiple tasks, rather than any specific heuristic or method to outperform baselines on one given setting. Our main contributions comprise:

- **Unified Framework:** We introduce **MIRA**, a framework that leverages biologically-inspired associative memories to propose a unified architecture for DG, CIL, and DIL.
- **Key Refinement of Hopfield Networks:** Our core technical novelty lies in embedding Hopfield networks in every ViT layer, dynamically aligning their keys to preceding layer activations instead of using static indexing keys, thus allowing the model to learn appropriate indexing rules.
- **Comprehensive empirical evaluation:** We study **MIRA** on standard benchmarks across multiple settings, attaining state-of-the-art (SoTA) accuracy in multiple settings, outperforming task-specialized architectures by as much as 10% in some cases.

## 2 Background and Related Work

**Memory in Deep Learning**: Traditional Hopfield networks [34] pioneered the computational models of associative memories by allowing a set of stored binary patterns to be retrieved via energy minimization. Recent variants, including Modern Continuous Hopfield Networks (**MCHN**) [69] and Universal Hopfield Networks (**UHN**) [63] improved upon the original to achieve exponentially greater storage capacity in addition to storing and retrieving real-valued vectors. Other forms of explicit memory have also been previously used in various architectures [100, 81, 28, 29], integrating explicit read/write operations to an external memory module to support long-range dependency handling, albeit each having its own distinct formulation. Recent works have also studied memory networks that can operate via Predictive Coding to better emulate biological memories [106, 82].

Recent advancements have explored the integration of associative memory mechanisms into diverse machine learning paradigms. Saliency-Guided Hidden Associative Replay (SHARC) [5] framework utilizes associative memory to store and replay salient data representations, enhancing retention of prior knowledge. [43] present a spiking neural network model that emulates associative memory functions for classifying neuroimaging data. [103] develop a neuromorphic computing framework that integrates global and local learning mechanisms, drawing inspiration from the brain's associative memory processes. These works only consider memories as a storage medium for data, independent of the main forward pass. [2] postulates that associative memories can store and retrieve neuromodulatory signals given the input context, achieving performance comparable to storage and retrieval of model weights from disk. However, it considers the AM as a disjoint module from the main neural network, posing it as akin to a biologically-plausible storage medium. We instead use AMs as an integral part of the forward pass, adjustable via backpropagation.

**Adapters in Contemporary Models**: The surge in large pre-trained models has led to methods that minimize the computational and storage overhead associated with fine-tuning. Techniques like LoRA [36] and its variants [47, 24, 107, 40, 105, 19] factorize weight updates into low-rank matrices, while Prefix Tuning [55], Adapter Layers [35], LayerNorm Tuning [108], and BitFit [7] similarly constrain the number of trainable parameters. These methods significantly reduce computational and storage overhead by constraining trainable parameters to minimal subsets or low-rank structures.

**Unified Frameworks for Multiple ML Settings**: Machine Learning literature tackled multiple paradigms like Domain Generalization (DG) [109], Continual Learning (CL) [90], and multi-task learning (MTL) [15] largely in isolation with few exceptions [49, 65]. Our work focuses on a unified framework driven by the practical necessity of robust models that generalize across new and previously unseen tasks, or learning new tasks and domains without forgetting prior knowledge. Unlike existing methods, our **MIRA** framework brings a fresh approach, providing both high adaptability and robust knowledge retention in a single extensible architecture, marking a significant innovation in handling diverse and evolving ML challenges.

## 3 Method

**Preliminaries and Notation.** Formally, a task $t \in [T]$ is defined by a dataset, $\mathcal{D}_t = \left\{ (x_i^{(t)}, y_i^{(t)}) \right\}_{i=1}^{N_t}$, sampled from a probability distribution $\mathcal{P}^{(t)}(\mathcal{X}^{(t)} \times \mathcal{Y}^{(t)})$. Here, $T$ denotes the total number of tasks, and $\mathcal{X}^{(t)}$, $\mathcal{Y}^{(t)}$, the domains of features and labels for task $t$ respectively.

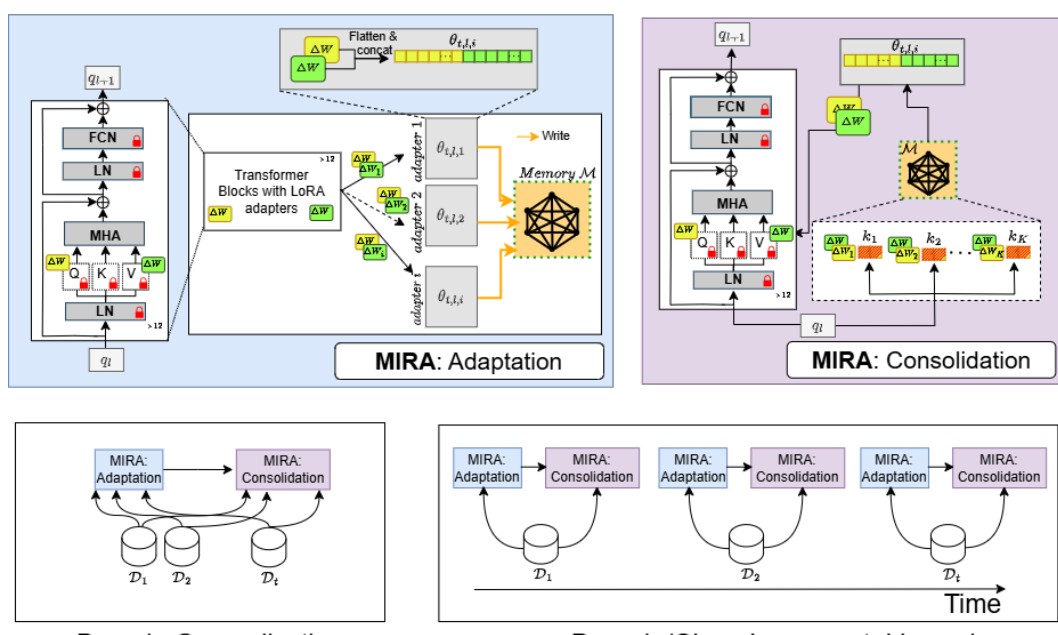

Figure 2: Overview of **MIRA** for Domain Generalization and Continual Learning scenarios. In DG, all training tasks are provided together to both the Adaptation and the Consolidation stages. In the CL scenarios, the dataset for each task arrives sequentially, and each dataset is passed to both stages. The Adaptation stage trains adapters for each task, while the Consolidation stage learns the associated keys for the stored adapters.

Our method is demonstrated over a frozen substrate network $\mathcal{F}$ (e.g., a ViT) consisting of $L$ layers. At each layer $\ell \in L$, we attach rank-$r$ LoRA adapters to the Query and Value matrices of the transformer, jointly represented by a flattened vector $\theta_\ell^{(t)} \in \mathbb{R}^{d_v}$.

Adapters are stored in **UHN** memory units $\mathcal{M}_\ell$ attached to each layer. Each memory unit is equipped with a *write* operation, denoted by $\mathsf{W}(\mathcal{M}_\ell, k, \theta)$, and a *read* operation $\mathsf{R}(\mathcal{M}_\ell, q)$. A *write* operation in the default **UHN** writes a key-value pair $(k, \theta)$ (in that order) to $\mathcal{M}_\ell$ and a *read* operation retrieves a weighted combination of values stored in $\mathcal{M}_\ell$ when prompted with a query $q$. Concretely, if $\mathbf{K}_\ell \in \mathbb{R}^{d_k \times N}$ and $\mathbf{\Theta}_\ell \in \mathbb{R}^{d_v \times N}$ represent $N$ keys-value pairs respectively:

$$\mathsf{R}(\mathcal{M}_\ell, q) \equiv \mathbf{\Theta}_\ell \, \mathsf{sep}\big(\mathsf{sim}(\mathbf{K}_\ell^\top, q)\big), \tag{1}$$

where sep is a separation function such as Softmax, and sim is a similarity function like the Euclidean inner product.

The Modern Hopfield Network [68] (MHN) was proposed with the Softmax function to achieve superlinear storage capacity with respect to the query embedding dimension. [63] generalized the notion proposed by the MHN to accommodate functions besides Softmax, dubbed "Separation functions" as these functions are responsible for assigning weights to stored memories during recall. Separation functions facilitate accurate retrieval of a specific memory by assigning a very high weight to the concerned memory, or allow for recall of a superposition of memories by assigning diffused weights. We primarily use an affine function to compute the weights of the combination.

We illustrate the efficacy of our approach on three learning paradigms comprising DG, DIL, CIL. The primary differences between these settings lies in how data is made available (summarized in Table 1), while the overarching objective in all these settings is to achieve high accuracy on the test distribution using optimally trained parameters $\varphi$ across all training tasks, i.e.:

$$\min_{\varphi} \mathbb{E}_{(x,y) \sim \mathcal{D}_{\text{test}}}[[\mathcal{F}(x; \varphi) \neq y]]. \tag{2}$$

**Conceptual Framework.** The chosen settings underscore the need for the trained parameters $\varphi$ to *consolidate* knowledge *adapted* from diverse tasks, enabling effective use of relevant information

| Data Availability | DG | DIL | CIL |
|---|---|---|---|
| Tasks arrive sequentially? | ✗ | ✓ | ✓ |
| Same label sets across tasks? | ✓ | ✓ | ✗ |
| Task identifier available at inference? | ✗ | ✗ | ✗ |
| Test distribution seen during training? | ✗ | ✓ | ✓ |

Table 1: Comparison of data availability across ML settings.

during inference. This is usually done via multiple "expert" models, each learning a subset of the tasks at hand, making their individual predictions [4, 14, 72]. Heuristics then dictate which output is considered at the end. At their core, however, all these methods rely on the parameters learned per task, which serve as reservoirs of task-specific knowledge acquired during training. Memory systems capable of storing and retrieving such parameters based on a (possibly transformed) input query are thus a natural fit in such settings, as they allow retrieval of input-specific weight combinations on demand. The central question, then, is how to efficiently retrieve an appropriate set or ensemble of parameters on a per-sample basis at test time. Formally, for an input dataset $\mathcal{D}$, this can be posed as an optimization problem over the weights of the combination, a.k.a:

$$\{\alpha_{t,l}^*\}^{T,L} := \arg\min_{\alpha_{t,l}} \mathbb{E}_{(x,y)\sim\mathcal{D}} \left[ \left[ \mathcal{F}\left(x; \sum_{t=1}^{T}\sum_{l=1}^{L}\alpha_{t,l}(x)\theta_{t,l}\right) \neq y \right] \right]. \tag{3}$$

In general the above problem is highly non-convex and may involve first/second order methods to solve, which is computationally demanding at a per-sample level. We critically observe that, for an arbitrary key matrix $M$, Euclidean inner product as the similarity function, and values $\Theta_\ell$, the associative memory retrieves consolidated parameters for $\mathcal{F}$ if $\alpha_{t,\ell} = \mathsf{sep}(\langle M, g_\ell(x)\rangle)$ for some function $g_\ell(x)$. The following lemma formalizes this intuition:

**Lemma 1.** *Let $\mathcal{H}_k$ denote the reproducing-kernel Hilbert space induced by the kernel $k(\cdot,\cdot)$, and assume an optimal solution to Eqn. 3 $\{\alpha_{t,l}^*(x)\}^{T,L}$ admits a representation in a finite eigenbasis of the integral operator associated with $k$. Then, for any dataset $\mathcal{D}$ drawn from a distribution encountered during training, **AM** retrieval allows obtaining the minimum characterized by Eqn. 3.*

*Proof.* By Mercer's theorem on compact spaces [64], $\mathcal{H}_k$ always has an orthonormal eigenbasis. Since we assume $\{\alpha_{t,l}^*(x)\}^{T,L}$ to admit a representation via some finite number of eigenfunctions $f_i$, $1 \leq d \leq q$, we can pick the function $g_\ell$ to be the function that outputs $[f_1(x), f_2(x)\ldots f_q(x)]^{\mathsf{T}}$, the key matrix $M$ as the corresponding eigenvalues. Since such a choice is subsumed under the **AM** retrieval dynamics elucidated in Equation 1, the assertion in the lemma follows. $\square$

Importantly, this retrieval operates per-sample at inference without requiring gradient computation. Thus, motivated by **AM** systems, and specifically **UHN**s, we propose a simple and general mechanism for retrieving adapter weights. Assuming the availability of appropriate keys indexing into the adapters, retrieving an effective ensemble reduces to computing inner products between the query $x_{\mathsf{test}}$ (or its processed representation) and the stored keys. Consequently, the adapter weight combination is dictated by the geometry of the inner product space in which the **UHN** operates. This strategy yields a single, task-agnostic model that dynamically composes per-task adapters via an **AM**. Conceptually, training proceeds in two distinct stages. The first involves standard training of independent adapters per task using any suitable method for the setting of interest, followed by storing these adapters into the associative memory. This storage requires a set of keys capable of indexing the adapters, which may be either fixed or randomly initialized. The second stage aims to ensure that the consolidated adapter weights retrieved via associative memory, when loaded into the backbone network $\mathcal{F}$, perform effectively on their corresponding tasks. If performance is suboptimal, the retrieval keys are refined to improve alignment between the retrieved adapter ensemble and the task it was originally trained for. Conceptually, this constitutes a constrained variant of the first stage, where updates are restricted to lie in the subspace spanned by the similarity-weighted combination of adapters learned earlier. Viewed differently, this stage attempts to solve Equation 3 over a training set where $y_{\mathsf{train}}$ is *known*. Crucially, it facilitates implicit consolidation of cross-task knowledge via retrieval dynamics of the shared **AM**. Unlike prior uses of associative memories for storing raw content such as images or feature representations [5, 106, 74], our formulation embeds the Hopfield network directly into the training loop. Moreover, standard strategies to mitigate catastrophic forgetting can be readily

incorporated, as the adapter combinations themselves are treated as trainable parameters. Finally, whereas the first stage trains separate adapters per task and layer $\ell$, the refinement of retrieval keys in stage two depends on the specific learning scenario: for **DG**, all domain data is jointly accessible and thus the partition is the whole dataset itself, whereas for continual learning settings, tasks are discarded sequentially, and so the partitions in this case are the individual tasks themselves.

---

**Algorithm 1: MIRA: Training**

---

**Require**: Tasks $\{\mathcal{D}_t\}_{t=1}^T$, frozen backbone $\mathcal{F}$, **AM** models $\bigcup_{\ell=1}^L \mathcal{M}_\ell$

1: **for** $t = 1$ **to** $T$ **do**
2:  Adaptation$(\mathcal{D}_t, \mathcal{F}, \bigcup_{\ell=1}^L \mathcal{M}) :=$
   $\begin{cases} \text{/* Train adapters } \theta_{t,\ell,i} \text{ via method specific loss. */} \\ \text{/* Write } \theta_{t,\ell,i} \mapsto \mathcal{M} \text{ via placeholder keys. */} \end{cases}$
3:   **if** Setting $==$ CL **then**
4:     Consolidation$(\mathcal{D}_t, \mathcal{F}, \bigcup_{l=1}^L \mathcal{M}_\ell) :=$
     $\begin{cases} \text{/* Finetune only keys via second pass over data. */} \\ \text{/* Apply CL heuristics to handle catastrophic forgetting. */} \end{cases}$
5:   **end if**
6: **end for**
7: **if** Setting $==$ DG **then**
8:   **for** $t = 1$ **to** $T$ **do**
9:     Consolidation$(\mathcal{D}_t, \mathcal{F}, \bigcup_{l=1}^L \mathcal{M}_\ell)$
10:   **end for**
11: **end if**

---

This two-stage blueprint is formally outlined in Algorithm 1.

**Method Design.** We now delve into specific design choices we make as we instantiate Algorithm 1. Note that both subroutines described below assume the task given as input, as allocating the right task to each subroutine is handled in Algorithm 1.

**Two-Stage Training.** The first of the two stages, dubbed *Adaptation*, involves training the base network to adapt to a single new task. We perform this adaptation by training LoRA-style adapters for each layer, using the provided dataset. Once trained, these adapters are then stored in the associative memory. Since such storage requires data in the form of key-value pairs, we initially randomly choose keys for each adapter by sampling from $\mathcal{N}(0, \sigma^2 I)$. Notably, our method is *not* constrained to storing LoRA adapters; infact, entire weight matrices can be stored into the associative memory.

---

**Algorithm 2: MIRA: Adaptation**

---

**Input:** Dataset $\mathcal{D}$, frozen backbone $\mathcal{F}$, **AM** models $\bigcup_{\ell=1}^L \mathcal{M}_\ell$, hyperparameter $\sigma^2$

1: **for** $(x, y) \in \mathcal{D}$ **do**
2:   Train memory adapters $\theta_\ell$ on $\mathcal{D}$ with Cross-Entropy loss, $\forall \ell \in L$
3: **end for**
4: **for** $l \in L$ **do**
5:   Sample $k_\ell \sim \mathcal{N}(0, \sigma^2 I)$
6:   W$(\mathcal{M}_\ell, k_\ell, \theta_\ell)$
7: **end for**

---

We choose LoRA adapters for simplicity and ease of implementation. To produce consolidated parameters for a given input, we explicitly train the randomly initialized keys using backpropagation in the second stage dubbed *Consolidation*, to yield adapter ensembles that minimize cross-entropy over the training set.

**Algorithm 3: MIRA: Consolidation**

---

**Input:** Dataset $\mathcal{D}$, backbone $\mathcal{F}$, **AM**s $\bigcup_{\ell=1}^{L} \mathcal{M}_\ell$, frozen $\bigcup_{\ell=1}^{L} \Theta_\ell$, initial keys $\bigcup_{\ell=1}^{L} \mathbf{K}_\ell$

1: **for** $(x, y) \in \mathcal{D}$ **do**
2:    **for** $\ell \in L$ **do**
3:       Compute layer inputs $h_{\ell-1}$ using $\theta_{\ell-1}$ `// `$h_0 \leftarrow x$
4:       $q_\ell \leftarrow g_\ell(h_{\ell-1})$ `// `$g_\ell$ `:query module for layer ` $\ell$
5:       Read $\hat{\theta}_\ell = \mathsf{R}(\mathcal{M}_\ell, \Pi_\ell)$
6:    **end for**
7:    Compute Cross-Entropy loss and back-propagate
8:    Update only $\bigcup_{\ell=1}^{L} \mathbf{K}_\ell$ and $\bigcup_{\ell=1}^{L} g_\ell$
9: **end for**

---

For a fixed value stored in the **UHN**, the keys pointing to the values are continuously updated across sequential task exposures. Specifically, a query input to the ViT backbone $\mathcal{F}$ passes sequentially through a stack of layers, each prepended with a lightweight query module $g_\ell : \mathbb{R}^{d_h} \to \mathbb{R}^{d_k}$. This module transforms the output $h_{\ell-1} \in \mathbb{R}^{d_h}$ of the previous layer into a query vector for layer $\ell$. The module can be instantiated as an identity map, a linear transformation, or a small neural network.

The transformed query $q_\ell = g_\ell(h_{\ell-1})$ is then matched against the keys at layer $l$ to compute similarity scores, which are normalized using their sum (as opposed to the norm of the sum) and used to weight the corresponding adapters in the associative memory. This weighted ensemble is loaded onto the current layer, and the modulated layer output becomes the input for the next layer, $h_{\ell+1}$.

Both the keys and the query modules are updated via backpropagation. The query module serves to align the geometry of the layer outputs with the key space, as these may naturally lie in different representational domains. Ultimately, the joint training of keys and query modules aims to produce the appropriate adapter combinations for inputs sampled from the task distribution encountered during training. In continual learning settings, where catastrophic forgetting is a concern, we incorporate standard mitigation techniques such as DualGPM [59] within this framework. During inference, the architecture follows the procedure in Algorithm 3, except that no parameter updates are performed.

## 4 Experiments and Results

We rigorously evaluate **MIRA** across the three scenarios - Class-incremental Learning (CIL), Domain-incremental Learning (DIL), and Domain Generalization (DG), on several benchmark datasets. Details regarding training protocols and hyperparameter tuning are provided comprehensively in the Appendix. Following PEGO [37], we make use of the ViT-B/16 architecture [16] initialized with CLIP [69] weights, and use LoRA with rank 4 for adapting to downstream tasks.

**Datasets.** For the CIL and DIL scenarios, we adhere to the established setup from [65], benchmarking primarily on the widely-used datasets: iDigits, CORe50, and DomainNet [66]. For CIL, we partition the dataset classes into five sequential tasks, each consisting of a mixture from all available domains. In contrast, the DIL setup constructs sequential tasks, each encompassing all classes from a single domain. Additionally, we expand our benchmarking to incorporate more recent and challenging datasets: ImageNet-R [32] split into both 5-task and 10-task scenarios in CIL, and the popular CDDB [53] dataset along with the recent DN4IL dataset [26] for the DIL setting. For the DG scenario, our evaluation covers four prominent datasets: PACS [54], VLCS [85], OfficeHome [87], and DomainNet.

**Baselines.** We benchmark **MIRA** against an extensive suite of SoTA baselines tailored specifically to each scenario. For the CIL and DIL settings, we include classical regularization-based approaches, such as Elastic Weight Consolidation (EWC) [46] and Learning without Forgetting (LwF) [57]. Moreover, we compare against cutting-edge parameter-efficient fine-tuning (PEFT) methods, including S-Prompts [39], L2P [98], DualPrompt [98], CODA-Prompt [93], and LAE [94]. We further compare to ICON [65], designed specifically for unified handling of CIL and DIL scenarios, though lacking inherent DG capabilities. For the DG scenario, we benchmark against state-of-the-art methods including popular methods like SWAD [50] and CoOP [110], and more recent works like PEGO [37].

**Evaluation Metrics.** We employ two principal metrics extensively utilized in incremental learning literature: *Average Accuracy* (Avg. Acc↑), where higher values indicate superior overall performance,

Table 2: Comparison with SoTA CIL and DIL methods on three standard datasets. Baseline numbers have been taken from prior work. In the CIL setting, we divide all classes into 5 distinct tasks, while in the DIL setting, every domain serves as a separate task. **MIRA** outperforms all baselines on average accuracy, with minimal forgetting. Best results are highlighted in **bold**, and results within 2% of the best are underlined.

| Dataset | Method | CIL | | DIL | | Avg. Acc |
| --- | --- | --- | --- | --- | --- | --- |
| | | Avg. Acc.↑ | Forgetting↓ | Avg. Acc.↑ | Forgetting↓ | |
| **iDigits** | Fine-tuning | 30.32±0.77 | 48.01±0.72 | 33.04±0.89 | 23.23±0.74 | 31.68 |
| | EWC [45] | 34.16±0.32 | 38.72±0.59 | 68.62±0.92 | 25.94±0.98 | 51.39 |
| | LwF [56] | 39.88±0.91 | 33.35±0.52 | 69.61±0.33 | 25.81±0.69 | 54.75 |
| | L2P [97] | 63.17±0.88 | 28.53±0.81 | 73.83±0.26 | 23.43±0.65 | 68.50 |
| | S-Prompts [91] | 55.09±3.27 | 25.61±1.62 | 75.11±2.31 | 25.66±6.23 | 65.10 |
| | DualPrompt [96] | 68.82±0.97 | 11.81±1.77 | 76.42±0.46 | 26.33±0.62 | 72.62 |
| | CODA-P [80] | 69.97±1.02 | 19.83±2.28 | 77.42±0.71 | 22.20±0.18 | 73.70 |
| | LAE [22] | 65.77±0.83 | 28.47±0.77 | 79.09±1.03 | 21.86±0.40 | 72.43 |
| | ICON [65] | 71.53±0.68 | 19.36±1.17 | **84.83±0.51** | 12.67±0.61 | 78.18 |
| | **Ours (MIRA)** | **83.00±1.29** | **10.62±2.80** | 82.46±0.12 | **8.49±0.43** | **82.73** |
| **CORe50** | Fine-tuning | 21.54±1.91 | 74.05±1.31 | 23.52±0.26 | 3.09±0.11 | 22.53 |
| | EWC [45] | 33.89±0.83 | 50.18±0.30 | 73.86±0.38 | 1.09±0.12 | 53.88 |
| | LwF [56] | 34.53±0.55 | 41.05±0.30 | 74.35±0.52 | 0.81±0.27 | 54.44 |
| | L2P [97] | 70.03±0.51 | 6.51±0.59 | 80.72±0.39 | 0.51±0.28 | 75.38 |
| | S-Prompts [91] | 68.27±3.92 | 11.79±0.24 | 86.50±0.46 | 0.92±0.31 | 77.39 |
| | DualPrompt [96] | 71.96±0.37 | 5.04±0.71 | 81.41±0.22 | 0.21±0.76 | 76.69 |
| | CODA-P [80] | 77.85±0.44 | **4.78±0.37** | 84.36±1.04 | 0.64±0.14 | 81.11 |
| | LAE [22] | 77.11±0.31 | 18.38±1.67 | 83.09±0.71 | 0.17±0.51 | 80.10 |
| | ICON [65] | 80.85±0.23 | 7.68±0.52 | 89.01±0.33 | 0.17±0.21 | 84.93 |
| | **Ours (MIRA)** | **83.39±0.24** | 7.99±1.43 | **93.89±0.33** | **0.00±0.00** | **88.64** |
| **DomainNet** | Fine-tuning | 35.43±0.58 | 47.79±0.28 | 39.52±0.32 | 28.81±0.64 | 37.48 |
| | EWC [45] | 53.04±0.53 | 24.41±0.48 | 41.58±0.26 | 26.79±0.15 | 47.31 |
| | LwF [56] | 53.79±0.61 | 19.41±0.11 | 43.74±0.27 | 18.23±0.10 | 48.77 |
| | L2P [97] | 60.90±0.69 | 8.23±0.90 | 48.55±0.81 | 19.71±1.29 | 54.73 |
| | S-Prompts [91] | 39.78±0.62 | 19.29±1.04 | 50.80±0.63 | 4.20±0.53 | 45.29 |
| | DualPrompt [96] | 62.55±0.92 | 7.62±1.07 | 51.33±0.10 | 9.60±1.41 | 56.94 |
| | CODA-P [80] | 65.21±0.24 | 15.01±0.21 | 49.13±0.83 | 25.96±1.13 | 57.17 |
| | LAE [22] | 65.06±0.18 | 9.68±0.84 | 44.67±0.62 | 28.99±0.64 | 54.87 |
| | ICON [65] | 65.43±0.15 | 9.72±0.46 | 54.44±0.21 | 13.32±0.46 | 59.94 |
| | **Ours (MIRA)** | **67.29±0.19** | **7.60±1.06** | **69.18±0.10** | **4.07±0.15** | **68.24** |

and *Forgetting*, where lower values imply better retention of previously learned tasks. We follow standard protocol [65, 96, 80] for reporting these metrics, emphasizing the final task accuracy after completing all incremental tasks.

**Main Results.** Table 2 summarizes the extensive experimental results in DIL and CIL scenarios. We utilize numbers reported from prior work where available to ensure fair comparison. **MIRA** consistently outperforms the state-of-the-art methods by a significant margin in both Avg. Accuracy and Forgetting metrics across these scenarios. For instance, on the iDigits dataset, **MIRA** achieves a notable Avg. Accuracy of 83% and a remarkably low Forgetting of just 10.62%, clearly surpassing the next best ICON by a significant margin. It should be noted that ICON was designed to handle both CIL and DIL tasks - **MIRA** achieves SoTA on both these settings, in addition to being suitable for DG. Similar trends are evident on CORe50 and DomainNet datasets, highlighting the robustness and effectiveness of our approach. We see similar trends in the case of the DG setting, with **MIRA** achieving SoTA performance on three out of four benchmark datasets, and being comparable to SoTA in the remaining dataset. The margin achieved by **MIRA** over baseline methods on the harder OfficeHome and DomainNet datasets are particularly significant. As in the case of CIL and DIL settings, the baseline methods were specifically proposed for DG, and are not directly applicable to Continual Learning settings.

**Additional Datasets.** To evaluate **MIRA** extensively, we employ additional benchmark datasets used by recent CIL and DIL works. In particular, in the DIL setting, we use the challenging CDDB-hard dataset, achieving SoTA performance as shown in Table 6. We also benchmark on the recently proposed DN4IL [26] dataset (Table 4), which currently lacks widespread use. We compare against methods that evaluate on it, and outperform them by a significant margin, setting a new SoTA baseline. Notably, we outperform methods that use a replay buffer with 200 exemplars, without using Exemplar

Table 3: Comparison with SoTA DG methods on 4 standard DG datasets. **Bold** = best; underlined = within 2% of best.

| Method | PACS | VLCS | OfficeHome | DomainNet | Avg |
|---|---|---|---|---|---|
| SWAD [11] | $91.30_{\pm 0.1}$ | $79.40_{\pm 0.4}$ | $76.90_{\pm 0.1}$ | $51.70_{\pm 0.8}$ | 74.33 |
| CLIP [67] | $96.20_{\pm 0.1}$ | $81.70_{\pm 0.1}$ | $82.00_{\pm 0.1}$ | $57.50_{\pm 0.1}$ | 79.85 |
| SMA [4] | $92.10_{\pm 0.2}$ | $79.70_{\pm 0.2}$ | $78.10_{\pm 0.1}$ | $55.90_{\pm 0.2}$ | 76.95 |
| ERM [86] | $93.70_{\pm 0.1}$ | $82.70_{\pm 0.1}$ | $78.50_{\pm 0.1}$ | $53.80_{\pm 0.1}$ | 77.68 |
| CoOp [111] | $96.20_{\pm 0.1}$ | $77.60_{\pm 0.2}$ | $83.90_{\pm 0.1}$ | $59.80_{\pm 0.1}$ | 79.88 |
| MIRO [12] | $95.60_{\pm 0.2}$ | $82.20_{\pm 0.2}$ | $82.50_{\pm 0.1}$ | $54.00_{\pm 0.3}$ | 78.58 |
| SEDGE [58] | $96.10_{\pm 0.1}$ | $82.20_{\pm 0.2}$ | $80.70_{\pm 0.2}$ | $54.70_{\pm 0.1}$ | 78.43 |
| GESTUR [52] | $96.00_{\pm 0.0}$ | $82.80_{\pm 0.1}$ | $84.20_{\pm 0.1}$ | $58.90_{\pm 0.1}$ | 80.48 |
| PEGO [37] | $96.50_{\pm 0.1}$ | $\mathbf{83.20_{\pm 0.3}}$ | $84.20_{\pm 0.1}$ | $57.30_{\pm 0.3}$ | 80.30 |
| **Ours (MIRA)** | $\mathbf{97.01_{\pm 0.0}}$ | $82.10_{\pm 0.5}$ | $\mathbf{87.36_{\pm 0.3}}$ | $\mathbf{61.19_{\pm 0.1}}$ | 81.92 |

Table 4: DN4IL (DIL, 200-exemplar buffer): single- vs multi-model methods.

| Method | Last Acc. ↑ |
|---|---|
| ER [70] | $27.45_{\pm 0.94}$ |
| DER++ [10] | $35.74_{\pm 0.67}$ |
| DARE [42] | $40.59_{\pm 0.73}$ |
| CLS-ER [3] | $41.70_{\pm 1.41}$ |
| DUCA [27] | $44.45_{\pm 0.18}$ |
| DARE++ [42] | $44.11_{\pm 0.98}$ |
| **Ours (MIRA)** | $\mathbf{78.40_{\pm 0.29}}$ |

Table 5: Comparison of recent SoTA methods on the Imagenet-R dataset in 5-task and 10-task CIL settings.

| Tasks | 5 | 10 |
|---|---|---|
| Method | $ACC_5$ (↑) | $ACC_{10}$ (↑) |
| Joint | $81.14 \pm 0.34$ | $81.14 \pm 0.34$ |
| Sequential | $58.74 \pm 1.28$ | $46.07 \pm 1.15$ |
| L2P [97] | $64.13 \pm 0.78$ | $62.54 \pm 0.24$ |
| DualPrompt [96] | $67.88 \pm 0.17$ | $65.41 \pm 0.52$ |
| CODA-P [80] | $73.09 \pm 0.21$ | $71.47 \pm 0.16$ |
| C-LoRA [79] | $75.85 \pm 0.31$ | $71.89 \pm 0.45$ |
| LAE [22] | $73.84 \pm 0.14$ | $71.70 \pm 0.39$ |
| **Ours (MIRA)** | $\mathbf{78.06 \pm 0.76}$ | $\mathbf{73.08 \pm 0.46}$ |

Table 6: Comparison of recent SoTA DIL methods on the CDDB dataset. 'Joint' refers to training on all experiences at once in a static setting instead of continually training, and serves as an upper bound on performance.

| Method | Average Acc (↑) |
|---|---|
| EWC [45] | 50.59 |
| LwF [56] | 60.94 |
| DyTox [17] | 51.27 |
| L2P [97] | 61.28 |
| S-iPrompts [92] | 74.51 |
| **Ours (MIRA)** | $\mathbf{77.37 \pm 0.21}$ |
| Joint | 85.50 |

Replay ourselves. To evaluate **MIRA** over longer learning timeframes, we also evaluate it in 5-task and 10-task splits of the ImageNet-R dataset, outperforming recent baselines as seen in Table 5.

**Effectiveness of Separation Functions.** In addition to Softmax, other memory models have proposed the use of various separation functions - Identity (classical Hopfield Networks [34]), Polynomial (Dense Associative Memories [48]), ReLU (proposed in [33]) and Linear (Tolman-Eichenbaum Machine (TAM) [101, 102], which attempts to model the Hippocampus).

We investigate the influence of different separation functions in our model architecture. Specifically, we explore variants employing affine transformation, Softmax normalization, and ReLU and Tanh activations. Note that affine and Tanh functions allow "removing" destructive information from the output representations by allowing negative weights to be assigned to certain adapters, while Softmax and ReLU can at most "mask" such information with zero or very low coefficients.

The affine variant (used in our primary experiments) intuitively allows flexible linear transformations, capturing nuanced task relationships, in addition to reflecting the TAM model. Alternatively, the Softmax variant ensures a probabilistic distribution over adapter activations, potentially beneficial in scenarios demanding explicit competition among adapters. The ReLU variant introduces sparsity, which might reduce interference by selectively activating only relevant adapters. Finally, the Tanh activation offers an alternative to the affine variant, while providing nearly equal weights for relevant adapters and allowing deteriorating information to be actively removed using negative coefficients.

Our results, as tabulated in Table 7, indicate that the ability to actively remove interfering information is key to the CIL setting, with the affine and Tanh variants achieving the best performance. On the other hand, allowing negative coefficients may result in removing relevant information, which may explain their reduced performance in the DIL setting. However, the DG setting clearly establishes that having a non-uniform selection of adapters, along with the ability to remove interfering information, holds the key to out-of-distribution generalization. We conclude that while all separation functions perform well, the affine function achieves the best performance overall.

**Impact of Adapter Count.** We conduct a detailed analysis examining the sensitivity of **MIRA** to the number of task-specific adapters employed (1, 2, 5, and 10 adapters per task). The results, tabulated in Table 8, indicate a clear improvement as the number of adapters per task or domain increases, highlighting the importance of capturing diverse task-specific nuances. However, increasing the adapter count beyond 5 yields marginal returns in performance improvement. Thus, our experiments suggest an optimal trade-off around five adapters per task, balancing efficiency and accuracy effectively. However, since efficiency is not the main concern of this work, we use 10 adapters per domain or task in all our experiments, since this gives us the best absolute results.

Table 7: Comparison of different separation functions used for retrieval from memory.

| Sep. Func | CIL Acc. | DIL Acc. | DG Acc. | Avg. |
|---|---|---|---|---|
| Affine | 67.29 | 69.18 | 61.19 | **65.89** |
| Softmax | 66.87 | 69.21 | 60.82 | 65.63 |
| ReLU | 66.60 | 69.20 | 60.90 | 65.57 |
| Tanh | 66.73 | 68.96 | 60.94 | 65.54 |

Table 8: Effects of number of adapters trained for each task/domain.

| #Adapters | CIL Acc. | DIL Acc. | DG Acc. | Avg. |
|---|---|---|---|---|
| 1 | 63.75 | 69.08 | 61.21 | 64.68 |
| 2 | 66.93 | 69.04 | 61.19 | 65.72 |
| 5 | 67.21 | 69.10 | 61.01 | 65.77 |
| 10 | 67.29 | 69.18 | 61.19 | 65.89 |

**Performance Overhead of Hopfield Keys.** We measure the inference-time latency and memory overhead introduced by trained integrated Hopfield keys (after Algorithm 3) on an adapters-loaded ViT backbone, relative to the same backbone without keys (i.e., using standard LoRA). On DomainNet-DIL with ViT-B/16 (LAION initialization), the average latency is $0.0241$s with keys vs. $0.0240$s without, i.e., a negligible $\sim 0.4\%$ overhead. Using the identity-based query module with 5 adapters per task across 6 tasks and key dimension 768 adds only $\sim 276$K parameters to an 86M-parameter model ($< 0.4\%$ increase in memory).

## 5 Conclusion

We presented **MIRA**, an architecture inspired by biologically plausible **AM**s, that unifies DG, DIL, and CIL scenarios. It uses an explicit **UHN** module for storing and retrieving low-rank weight adapters. Empirically, **MIRA** delivers SoTA performance across all three settings, often surpassing specialized baselines by significant margins, while requiring only minor objective tweaks rather than entire architectural changes. Beyond raw accuracy, our results demonstrate that coupling deep networks with neuro-plausible memory mechanisms yields flexible, reusable circuitry that can continually incorporate new information and adapt swiftly to distributional shifts. We believe **MIRA** marks a step toward closing the gap between neural task-switching in biology and continual adaptation in artificial systems, and opens a rich avenue for exploring memory-centric, learning paradigms at scale, from vision to multimodal generative models.

## 6 Acknowledgment and Funding Transparency Statement

Susmit Agrawal was supported by the German Research Foundation (DFG): SFB 1233, Robust Vision: Inference Principles and Neural Mechanisms, TP C2, project number: 276693517, and the Reliance Postgraduate Fellowship. Susmit Agrawal thanks the International Max Planck Research School for Intelligent Systems (IMPRS-IS) and the Reliance Foundation for support. Krishn V. Kher thanks the Prime Minister's Research Fellowship (PMRF) for funding and support. Vineeth N. Balasubramanian thanks Microsoft Research India, Bangalore for funding and support. All authors thank Mr. Aveen Dayal for guidance on choosing datasets for running experiments on the Domain Generalization setting. All authors thank the Indian Institute of Technology, Hyderabad for providing the resources and infrastructure required to develop this project.

# A Appendix

**Contents**

## A.1 MIRA: A Multi-Perspective Analysis

Our method, **MIRA** is a generic framework that can be viewed from four contemporary, broad perspectives in the general context of machine learning, going beyond the perspective introduced in the main paper. We elaborate on those perspectives below.

### A.1.1 Meta-learning (Hypernetworks)

Hypernetworks are models that predict weights of other models [30, 13, 6]. They offer a single-loop approach to meta-learning by eliminating the need for an inner-loop adaptation (as in MAML [20]), instead learning to output task-specific parameters in one forward pass. Since **MIRA** predicts adapters at each consecutive block hierarchically, the associative memories in **MIRA**'s architecture can be viewed as a specific type of hypernetwork. However, hypernetwork training is often unstable and is unable to support large models such as ViT architectures. We address this issue by providing a "supervision" to guide the direction of learning of the underlying hypernetwork. Hypernetworks then essentially become models that map a deterministic set of inputs to their outputs deterministically, resembling the behavior of associative memories, and can be implemented as such. We posit this perspective in this work, wherein weight retrieval is achieved by storing the adapter weights over tasks defined over the training set, and retrieving them differentiably, instead of attempting to both learn and memorize them simultaneously as done in traditional hypernetwork-based learning. Thus, Hopfield Networks [69], Predictive Coding Networks [106, 83], and any such **AM** may in theory be used in this framework. Practical experiments, however, indicated that despite PCNs [106] exhibiting better compression properties than Hopfield Nets, they usually have unsatisfactory retrieval quality when the vectors to be retrieved have very high dimension. This is expected behavior, especially in the context of storing weight adapters of large foundational models. In addition, their reliance on algorithms incompatible with backpropagation makes it difficult to integrate them into models that need to be trained end-to-end. Hopfield Nets, on the other hand, provide high fidelity in retrieving such high-dimensional vectors at scale, and are implicitly differentiable.

### A.1.2 Functional Interpolation/Extrapolation

In this work, we utilize affine combinations of task-specific adapters for retrieval across different domains. However, our associative memory-centric learned retrieval framework is versatile and can seamlessly accommodate richer, non-linear retrieval mechanisms by modifying the underlying similarity metric strategy. In principle, one could design much more expressive (non-linear) combination schemes to merge knowledge from multiple domains, rather than restricting to linear interpolation [112]. This is an interesting direction of future work for this paper. Such non-linear retrieval approaches have been explored in recent literature, such as in sparsely-gated mixture-of-experts models [78] use a learned gating function to dynamically select only a subset of expert parameters for each input (instead of a fixed weighted average) [31], or in using learned retrieval functions (e.g., trainable

hashing or routing instead of standard nearest-neighbor search) yields better scaling and can capture latent structure in the memory, outperforming fixed similarity measures [31]. Crucially, there is both practical and theoretical evidence of directly retrieving such non-linear ensembles of experts/adapters in Hopfield Networks, such as in [76].

Thus, our method can be viewed as retrieving an interpolated task-specific knowledge proxy (adapter) in a space defined by the chosen functional form of combination. With a sufficiently expressive interpolation function, this approach can even enable extrapolation to out-of-distribution tasks. In settings like **DG**, the target domain may lie outside the convex hull of the source domains, wherein the model must generalize beyond any seen domain mixture. A suitably rich, non-linear combination strategy could, in principle, facilitate extrapolation of the memorized adapter weights to novel task distributions [1]. Our approach thus offers a generalizable framework capable of both capturing nuanced relationships between known tasks and extending learned knowledge to new domains beyond the scope of training distributions. We leave the exploration of such function schemes to future work.

### A.1.3 Test Time Adaptation

At test time, the optimal combination of adapters for a given input may not be obtained using pre-defined weight coefficients. In general, determining the adapter coefficients that best serve a new downstream sample can require solving an optimization problem on a per-sample basis, as posited in Equation 3. In other words, Equation 3 formalizes the idea that the best adapter composition $\{\alpha^*_{t,l}(x)\}^{T,L}$ for an input $x$ is obtained by minimizing a suitable objective for that specific sample at inference time (instead of applying a fixed combination rule). This perspective aligns with the paradigm of test-time adaptation in literature, wherein models trained only on source data are adapted to target data during inference time [104, 41]. As an example, Tent (Fully Test-Time Entropy Minimization) performs online model updates during testing by minimizing the entropy of its predictions for each test batch, thereby adjusting normalization parameters to increase the model's confidence on the target distribution [88]. One could view our **MIRA** framework as implementing this idea via a memory-based inference mechanism. Since **MIRA** learns a set of key representations (i.e., associative memory slots in a Hopfield network) during training that are used to derive weights on a per-sample basis, it can also use the learned keys at test time on a per-sample basis effectively, performing adaptation via associative recall. By casting test-time adaptation as an integral part of inference (through solving a Hopfield memory retrieval optimization akin to Equation 3), **MIRA** can be viewed also as a test-time adaptation strategy within a unified, optimized memory-based framework.

### A.1.4 Biological Perspective

Notably, **MIRA** can also be perceived as a biologically plausible framework that solves multiple settings such as **DG**, **CIL**, and **DIL**. While Hopfield Nets are well known as biologically implementable **AM** mechanisms, even the affine combinations that we adopt are well-founded in biological mechanisms. Specifically, Tolman-Eichenbaum Machine (TAM) [101, 102], a model of the Hippocampus, proposes linear combinations of stored memories as implementable (illustrated in Section A.2). We further observe that the incremental learning settings, when accompanied by DualGPM [60] as the strategy to mitigate catastrophic forgetting, resolve into Generalized Hebbian learning principles [75] in the gradient space of stored memories.

**Lemma 2** (DualGPM–Hebbian Gradient Subspace Equivalence). *Let $\mathcal{G}_t = \{g_i\}_{i=1}^{N_t} \subset \mathbb{R}^d$ be the set of gradient vectors observed while training on task $t$ and let the empirical second–moment matrix be*

$$\Sigma_t \;=\; \frac{1}{N_t}\sum_{i=1}^{N_t} g_i g_i^\top.$$

*In the* DualGPM *algorithm, for a given energy budget $\varepsilon \in (0,1)$, we define:*

$$k := \arg\min_{k\in[d]} \frac{\sum_{j=1}^{k}\lambda_j}{\sum_{j=1}^{d}\lambda_j} \;\geq\; \varepsilon.$$

*where $\lambda_1 \geq \cdots \geq \lambda_d$ are the eigenvalues of $\Sigma_t$.*

*Let $U_t \in \mathbb{R}^{d\times k}$ be the orthonormal basis produced by the* DualGPM *memory update for the energy budget $\varepsilon$. Independently, let $W_t \in \mathbb{R}^{d\times k}$ be the weight matrix obtained as a stationary point of the Generalised Hebbian (Oja) update averaged over each gradient in $\mathcal{G}_t$ (with row sums equal to 1).*

*Then,*

$$\mathrm{span}(U_t) = \mathrm{span}(W_t).$$

*Proof.* **Step 1: Optimality of DualGPM.** We first note that the precise objective that DualGPM tries to solve is given by:

$$U_t = \arg \min_{U^\top U = I_k} \mathrm{Tr}\big[(I - UU^\top)\Sigma_t\big]$$

By the Eckart-Young-Mirsky theorem, the optima is attained at those $U$, whose columns are the $k$ eigenvectors (subject to permutation and sign inversion) of $\Sigma_t$ corresponding to the $k$ *largest* eigenvalues, $\lambda_1, \ldots, \lambda_k$. Hence, $U_t$ satisfies the eigenvalue equation

$$\Sigma_t U_t = U_t \Lambda, \qquad \Lambda = \mathrm{diag}(\lambda_1, \ldots, \lambda_k). \tag{4}$$

**Step 2: Fixed points of the Hebbian rule.** The Generalized Hebbian update rule [**?** ] for a gradient vector $g \in \mathcal{G}_t$ is given by:

$$\Delta W(g) = \eta\big(gg^\top W - W \, \mathrm{diag}(W^\top gg^\top W)\big).$$

Considering the weight update averaged over all $g \in \mathcal{G}_t$ and imposing stationary conditions on the same yields:

$$\Sigma_t W - W \, \mathrm{diag}(W^\top \Sigma_t W) = 0.$$

Pre-multiplying by $W_t^\top$ and noting that each row sum of $W_t^\top$ equals one, shows that the diagonal term on the right is precisely the eigenvalue matrix of the projected covariance, so the above is identical to (4) with $U_t$ replaced by $W_t$.

Consequently, $U_t$ and $W_t$ have the same $k$-dimensional eigenbasis, corresponding to the $k$ eigenvectors of $\Sigma_t$ with the $k$-largest eigenvalues, which proves the assertion. $\square$

Thus, Lemma 2 implies that there exists an orthogonal matrix $R_{k \times k}$ such that $U_t = W_t R$.

**Implications.** This lemma elevates the biological analogy behind DualGPM into a provable equivalence: the algorithm's batch SVD update computes *exactly* the same principal gradient subspace that an online Oja-style Hebbian learner would converge to. The result has three immediate consequences: (i) *Theoretical grounding*: Any optimal variance or noise filtering guarantees enjoyed by Hebbian PCA now apply to DualGPM's memory, providing a principled basis for its strong empirical resistance to catastrophic forgetting; (ii) *Algorithmic unification*: Projection-based continual learning methods can be re-interpreted through the lens of gradient-space Hebbian consolidation; and (iii) *Neuro-inspired design*: By demonstrating that protecting past tasks is tantamount to a Hebbian consolidation step, the lemma bridges continual learning research with synaptic consolidation theories in neuroscience, motivating biologically grounded extensions such as local online updates or neural gating via associative memory.

## A.2 Additional Experiments

**Multiple Initializations and Backbones.** For completeness, Table A1 reports **DIL** results for **MIRA** across architectures and initializations. In addition to the ViT-B/16 backbone used in the main paper, we include ViT-B/32, whose performance is generally lower, reflecting the sensitivity of Hopfield components to both initialization and backbone choice. We also report results with the ViT-in21k initialization alongside the LAION initialization used in the main results.

Comparable experiments for the **DG** setting appear in Table A2. Notably, using ViT-in21k, we compare **MIRA** against a bare ViT-B/16 backbone with the same initialization but without adapters, and find that adapters with learned Hopfield keys yield substantial gains even over such strong baselines.

**Choice of $g()$.** In the tables in the main paper, we set $g$ described in the conceptual framework to be an identity function. We tabulate the results below for different choices of $g$. We find that this function has minimal impact on performance, indicating that the transformations performed within the different layers of ViT are strong and fairly sufficient by themselves to constitute an eigenbasis wherein the appropriate keys can be found via gradient descent.

Table A1: Comparison with SoTA DIL methods on three standard datasets. Baseline numbers have been taken from prior work. **MIRA** is evaluated on multiple ViT backbones and initializations for completeness. Best results are highlighted in **bold**, and results within 2% of the best are underlined.

| Dataset | Method | DIL | | Avg. Acc |
|---|---|---|---|---|
| | | Avg. Acc.↑ | Forgetting↓ | |
| iDigits | Fine-tuning | 33.04±0.89 | 23.23±0.74 | 31.68 |
| | EWC [45] | 68.62±0.92 | 25.94±0.98 | 51.39 |
| | LwF [56] | 69.61±0.33 | 25.81±0.69 | 54.75 |
| | L2P [97] | 73.83±0.26 | 23.43±0.65 | 68.50 |
| | S-Prompts [91] | 75.11±2.31 | 25.66±6.23 | 65.10 |
| | DualPrompt [96] | 76.42±0.46 | 26.33±0.62 | 72.62 |
| | CODA-P [80] | 77.42±0.71 | 22.20±0.18 | 73.70 |
| | LAE [22] | 79.09±1.03 | 21.86±0.40 | 72.43 |
| | ICON [65] | **84.83±0.51** | 12.67±0.61 | 78.18 |
| | **Main (MIRA)** | 82.46±0.12 | **8.49±0.43** | 82.46 |
| | **ViT-B/32 (MIRA)** | 83.06±0.12 | 18.42±0.43 | **83.06** |
| CORe50 | Fine-tuning | 23.52±0.26 | 3.09±0.11 | 22.53 |
| | EWC [45] | 73.86±0.38 | 1.09±0.12 | 53.88 |
| | LwF [56] | 74.35±0.52 | 0.81±0.27 | 54.44 |
| | L2P [97] | 80.72±0.39 | 0.51±0.28 | 75.38 |
| | S-Prompts [91] | 86.50±0.46 | 0.92±0.31 | 77.39 |
| | DualPrompt [96] | 81.41±0.22 | 0.21±0.76 | 76.69 |
| | CODA-P [80] | 84.36±1.04 | 0.64±0.14 | 81.11 |
| | LAE [22] | 83.09±0.71 | 0.17±0.51 | 80.10 |
| | ICON [65] | 89.01±0.33 | 0.17±0.21 | 84.93 |
| | **Main (MIRA)** | **93.89±0.33** | **0.00±0.00** | **93.89** |
| | **ViT-B/32 (MIRA)** | 91.28±0.33 | **0.00±0.00** | 91.28 |
| DomainNet | Fine-tuning | 39.52±0.32 | 28.81±0.64 | 37.48 |
| | EWC [45] | 41.58±0.26 | 26.79±0.15 | 47.31 |
| | LwF [56] | 43.74±0.27 | 18.23±0.10 | 48.77 |
| | L2P [97] | 48.55±0.81 | 19.71±1.29 | 54.73 |
| | S-Prompts [91] | 50.80±0.63 | 4.20±0.53 | 45.29 |
| | DualPrompt [96] | 51.33±0.10 | 9.60±1.41 | 56.94 |
| | CODA-P [80] | 49.13±0.83 | 25.96±1.13 | 57.17 |
| | LAE [22] | 44.67±0.62 | 28.99±0.64 | 54.87 |
| | ICON [65] | 54.44±0.21 | 13.32±0.46 | 59.94 |
| | **Main (MIRA)** | **69.18±0.10** | **4.07±0.15** | **69.18** |
| | **ViT-in21k (MIRA)** | 59.44±0.10 | 12.06±0.15 | 59.44 |
| | **ViT-B/32 (MIRA)** | 59.36±0.10 | 11.60±0.2 | 59.36 |

**Prefixes as memories.** The proposed **MIRA** framework bears resemblance to the complementary learning system implemented by hippocampal-cortical connections in the brain [77]. This framework, however, models the hippocampus as a memory storage for representations, rather than neural overlays. The analogue to such a mechanism in contemporary deep learning architectures is Prompt Tuning [98] in PEFT literature. In particular, the prefix tuning [55] variant of prompt tuning can be directly integrated into the **MIRA** framework to implement a system analogous to a neuroscientific framework, with the pretrained network serving the role of the cortical circuits with powerful generalization capabilities, and the prefixes - stored in associative memories - serving as task-specific representations. We compare the prefix-tuning based approach with our original **MIRA** framework. Our results indicate that this variant maintains comparable performance to storing overlay weights, showing that **MIRA** can be adapted to different PEFT methods.

## A.3 Experimental Details

For training task-specific adapters in the *Adaptation* stage across all datasets and settings, we use rank-4 LoRA adapters trained for 5 epochs with a learning rate of 1e-3. For CIL and DIL experiments, we set the DualGPM threshold to 0.7. The AdamW optimizer is used with a weight decay of 1e-3 across all experiments as well. All our experiments are performed on a single RTX A6000 Ada GPU with 48GB VRAM, on a machine having a 96-core Intel Xeon CPU and 128GB RAM.

In the *Consolidation* stage, all experiments in DIL and CIL settings ran for 2 epochs. In addition, we initialized the CIL classifiers in the *Consolidation* stage with the weights learned in the *Adaptation*

Table A2: Comparison with SoTA DG methods on 4 standard DG datasets with different initializations and ViT backbones. **Bold** = best; underlined = within 2% of best.

| Method | PACS | VLCS | OfficeHome | DomainNet | Avg |
|---|---|---|---|---|---|
| SWAD [11] | $91.30_{\pm 0.1}$ | $79.40_{\pm 0.4}$ | $76.90_{\pm 0.1}$ | $51.70_{\pm 0.8}$ | 74.33 |
| CLIP [67] | $96.20_{\pm 0.1}$ | $81.70_{\pm 0.1}$ | $82.00_{\pm 0.1}$ | $57.50_{\pm 0.1}$ | 79.85 |
| SMA [4] | $92.10_{\pm 0.2}$ | $79.70_{\pm 0.2}$ | $78.10_{\pm 0.1}$ | $55.90_{\pm 0.2}$ | 76.95 |
| ERM [86] | $93.70_{\pm 0.1}$ | $82.70_{\pm 0.1}$ | $78.50_{\pm 0.1}$ | $53.80_{\pm 0.1}$ | 77.68 |
| CoOp [111] | $96.20_{\pm 0.1}$ | $77.60_{\pm 0.2}$ | $83.90_{\pm 0.1}$ | $59.80_{\pm 0.1}$ | 79.88 |
| MIRO [12] | $95.60_{\pm 0.2}$ | $82.20_{\pm 0.2}$ | $82.50_{\pm 0.1}$ | $54.00_{\pm 0.3}$ | 78.58 |
| SEDGE [58] | $96.10_{\pm 0.1}$ | $82.20_{\pm 0.2}$ | $80.70_{\pm 0.2}$ | $54.70_{\pm 0.1}$ | 78.43 |
| GESTUR [52] | $96.00_{\pm 0.0}$ | $82.80_{\pm 0.1}$ | $84.20_{\pm 0.1}$ | $58.90_{\pm 0.1}$ | 80.48 |
| PEGO [37] | $96.50_{\pm 0.1}$ | $\mathbf{83.20_{\pm 0.3}}$ | $84.20_{\pm 0.1}$ | $57.30_{\pm 0.3}$ | 80.30 |
| **Main (MIRA)** | $\mathbf{97.01_{\pm 0.0}}$ | $82.10_{\pm 0.5}$ | $\mathbf{87.36_{\pm 0.3}}$ | $\mathbf{61.19_{\pm 0.1}}$ | **81.92** |
| **ViT-in21k (Base model)** | $68.89_{\pm 0.0}$ | $73.00_{\pm 0.5}$ | $81.23_{\pm 0.3}$ | $42.35_{\pm 0.1}$ | 66.36 |
| **ViT-in21k (MIRA)** | $71.28_{\pm 0.0}$ | $74.18_{\pm 0.5}$ | $82.52_{\pm 0.3}$ | $46.99_{\pm 0.1}$ | 68.74 |
| **ViT-B/32 (MIRA)** | $94.11_{\pm 0.0}$ | $81.68_{\pm 0.5}$ | $79.60_{\pm 0.3}$ | $53.07_{\pm 0.1}$ | 77.11 |

Table A3: Comparison of different choices of $g$ for DIL and DG settings.

| $g$ | DIL | DG |
|---|---|---|
| Identity | 69.18 | 61.19 |
| Linear | 69.22 | 60.98 |
| 3-layer MLP | 69.22 | 61.12 |

Table A4: Comparison of Prefix Tuning vs LoRA tuning in **MIRA**.

| MIRA Variant | DIL | DG |
|---|---|---|
| **MIRA**-default | 69.18 | 61.19 |
| **MIRA**-Prefixes | 69.61 | 60.72 |

stage for the corresponding label set. Note that this is not effective in the DIL setting, as even though the label sets are the same, the distribution of inputs to the classifier changes, and hence the scope of knowledge transfer in the linear classifier head is limited in this setting. In the *Consolidation* stage of the DG setting, we run PACS, OfficeHome, and VLCS for 10 epochs, while DomainNet is just run for one epoch. We rescale all images to $256 \times 256$ during both training and evaluation and take a $224 \times 224$ crop from this rescaled image (random crop during training, center crop at inference) as input to the model. We apply a random horizontal flip as a training augmentation in all cases, and an additional mixup augmentation for the DG setting. In all CIL and DIL settings, we use the AdamW optimizer with a weight decay and learning rate of 1e-3, while in the DG setting, we set the learning rate to 7e-4.

## A.4 Dataset Details

**DomainNet.** DomainNet is a large-scale benchmark comprising approximately 600,000 images across 345 categories, distributed over six distinct domains: Real, Clipart, Infograph, Painting, Quickdraw, and Sketch. Each domain introduces a unique visual style, presenting significant domain shifts. In the DIL setup, each domain is treated as a separate experience, with the model sequentially exposed to data from one domain at a time while maintaining a consistent label space. This setup challenges models to generalize across diverse visual domains without forgetting previously learned knowledge. In the CIL setup, the dataset is divided into 5 experiences, each experience containing 69 classes from all 6 domains combined. Unlike the DIL setting, the label space in the CIL setting grows with each experience. In the DG setup, models are trained on 5 domains conjointly and evaluated on the 6th unseen domain.

**DN4IL.** DN4IL is a curated subset of DomainNet, specifically designed for evaluating domain-incremental learning methods. It retains the six domains from DomainNet but focuses on a reduced

set of 100 classes to facilitate controlled experiments on domain shifts. The dataset emphasizes the challenges posed by significant distributional differences between domains, making it a suitable benchmark for assessing the robustness of continual learning algorithms.

**iDigits.** iDigits is a domain-incremental benchmark constructed by combining four digit recognition datasets: MNIST, SVHN, MNIST-M, and SYN. Each dataset represents a distinct domain with varying visual characteristics. In the DIL setting, the model is trained sequentially on each domain to maintain performance across all domains despite the domain shifts. This benchmark is particularly useful for studying the effects of domain shifts in simpler classification tasks. In the CIL setting, all datasets are jointly split into 5 training experiences, each experience containing 2 classes from each of the 4 datasets.

**CORe50.** CORe50 is a dataset designed for continuous object recognition, consisting of 50 household objects recorded under 11 different environmental conditions. Each condition introduces variations such as background changes, lighting, and occlusions. In the domain-incremental setup, each environmental condition is treated as a separate domain, and the model learns to recognize the same set of objects across these varying conditions. A key difference from other DIL datasets is that CORe50 uses 3 of the 11 domains as the test set and incrementally trains on the other 8 domains. This setup evaluates a model's ability to generalize object recognition across different real-world scenarios. It can thus also be viewed as a combination of DIL and DG settings, where the test set comprises of unseen domains. A forgetting of $\leq 0$ indicates that the models' performance remains the same or improves on the unseen domains as new domains are incrementally learned. The CIL setting is similar to DomainNet and iDigits - the dataset is split into 5 experiences of 10 classes each, encompassing all 11 training domains.

**CDDB.** CDDB (Continual Deepfake Detection Benchmark) is a dataset aimed at evaluating continual learning methods in the context of deepfake detection. It comprises a collection of deepfake videos generated using various known and unknown generative models. In the DIL framework, each generative model represents a different domain, and the model is sequentially trained to detect deepfakes from these diverse sources. CDDB challenges models to adapt to new types of deepfakes while retaining the ability to detect previously encountered ones. We particularly evaluate on the CDDB-hard subset, comprising five domains: GauGAN, BigGAN, WildDeepfake, WhichFaceReal, and SAN.

**ImageNet-R.** ImageNet-R is a dataset comprising 30,000 images of 200 ImageNet classes, with images rendered in various styles such as art, cartoons, graffiti, embroidery, and video games. This dataset is designed to evaluate the robustness of models to distribution shifts. In the CIL setup, the 200 classes are divided into 5 or 10 tasks, each containing 40 or 20 unique classes. The model is trained sequentially on these tasks to learn new classes while maintaining performance on previously learned ones.

**VLCS.** VLCS is a benchmark dataset for domain generalization, comprising images from four distinct domains: PASCAL VOC2007, LabelMe, Caltech-101, and SUN09. Each domain contains images labeled across five shared object categories: bird, car, chair, dog, and person. The dataset includes a total of 7,510 images, with domain-specific distributions. In the DG setup, models are trained on three domains and tested on the remaining one, evaluating their ability to generalize to unseen domains.

**PACS.** PACS is an image dataset designed for domain generalization, consisting of four domains: Photo, Art Painting, Cartoon, and Sketch. Each domain contains images from seven categories: dog, elephant, giraffe, guitar, horse, house, and person. The dataset comprises a total of 9,991 images, with varying numbers across domains. PACS introduces significant domain shifts due to the diverse visual styles, making it a challenging benchmark for DG methods.

**OfficeHome.** OfficeHome is a benchmark dataset for domain adaptation and generalization, containing images from four domains: Art, Clipart, Product, and Real-World. Each domain includes 65 categories of everyday objects, totaling approximately 15,500 images. The dataset presents substantial domain shifts due to differences in image styles and acquisition methods. In the DG setup, models are trained on three domains and evaluated on the fourth, assessing their ability to generalize to unseen domains.

### A.5 Theoretical Underpinnings

Lemma 1 stated that the **MIRA**'s **AM** formulation accommodates the optimal solution for Equation 3, as long as the optimal coefficients come from a kernel. It, however, does not prove that **MIRA** attains those optimal coefficients, specifically via the Consolidation phase of the training. Under certain mild assumptions on the kernel detailed below, we show that our method indeed converges to the true optimal coefficients. Our proof strategy largely follows [84], where, for convenience we also assume that the similarity function is softmax instead of an affine function as we have stated in the main text.

**Theorem 1.** *Let $\mathcal{H}_k$ denote the reproducing-kernel Hilbert space induced by the kernel $k(\cdot, \cdot)$, and assume an optimal solution to Eqn. 3 $\{\alpha_{t,l}^*(x)\}^{T,L}$ admits a representation in a finite eigenbasis of the integral operator associated with $k$. Further along the lines of **Att-SVM** in [84], define*

$$W(x) = K \cdot \sum_{t=1,l=1}^{T,L} \alpha_{t,l}^*(x) Q_{t,l}^T. \tag{5}$$

*Assume that the kernel $k$ is such that the following condition is true:*

$$\{\alpha_{t,l}^*(x)\}^{T,L} = \arg\min_{\alpha} \|W(x)\|, \ \ s.t. \ (x_{i_{opt_i}} - x_{it})W^T x_{i1} \geq 1, \ \forall t \neq opt_i, i \in [n]. \tag{6}$$

*Then the Consolidation stage of **MIRA** induces the query modules and learnable keys to converge to the optimal coefficients, $\alpha_{t,l}^*(x) Q_{t,l}^T$.*

*Proof.* The proof follows from [84], by replacing the matrix $W$ by our ensemble of adapters and inheriting their assumptions as is, specifically in Lemmas 1, 2, 4, 12, and Theorem 4. □

### A.6 Limitations

This work highlights the benefits of incorporating neuroscientific insights into deep learning architectures, especially in the context of biologically plausible memory mechanisms. In particular, it proposes a potential mechanism in which such task-switching can occur in biological systems with the aid of associative memories. The work constraints to task settings such as CIL, DIL, and DG; extensions to related settings such as Versatile Incremental Learning or Multi-Task Learning, or even other PEFT methods, would be interesting future extensions of our framework. All experiments provided are based on computational models from deep learning research; analogous neuroscience experiments may need to be conducted to conclusively declare if memory mechanisms are indeed used in the stated manner in biological systems. Besides, validating this framework on non-ViT architectures such as ResNets is also possible, and may help extend this work more generally to all architectures.

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
