# OpenReview forum: "Memory-Integrated Reconfigurable Adapters: A Unified Framework for Settings with Multiple Tasks"
_NeurIPS.cc/2025/Conference — NeurIPS 2025 poster_

### Official Review · Reviewer_C7LV · 2025-06-05

**Clarity:** 2
**Significance:** 2
**Originality:** 3
**Rating:** 3
**Confidence:** 3

**Summary:**

The paper proposes a unified framework using Hopfield associative memory (AM) with reconfigurable adapters to address Domain Generalization (DG), Domain-Incremental Learning (DIL), and Class-Incremental Learning (CIL). A fixed ViT backbone is supplemented with task-specific adapters stored as "values" in AM. Dynamically learned "keys" enable sample-based retrieval of relevant adapters during inference. Training involves adaptation and consolidation stages. Evaluation is performed on both domain generalization and continual learning tasks, and demonstrate the superiority of the proposed method.

**Questions:**

The two-stage training pipeline is a distinctive feature of the thesis. However, it would be valuable to elucidate whether this approach was essential or if it conferred particular benefits. In my view, it seems more like a limitation. Could the keys and memories be learned concurrently? For instance, Mixture of Experts (MoE) models do not necessitate two-stage learning.

**Ethical Concerns:**

["NO or VERY MINOR ethics concerns only"]

**Final Justification:**

After carefully reviewing the rebuttals and feedback from other reviewers, I will maintain my current evaluation.

First, while the authors assert that revisions have been incorporated in response to my concerns, I regret that I could not locate these additions in the revised manuscript. Clarity on where specific revisions address my points would be helpful.

Second, regarding the technical contributions: The memory-key-value design and two-stage optimization approach, while technically sound, appear to align closely with established mechanisms in frameworks such as self-attention and Mixture-of-Experts (MoE).  As the other reviewers has pointed out, It needs some more analyses on both training and testing efficiency, and practicality analyses.

**Quality:**

3

**Strengths And Weaknesses:**

**Strengths**
- First integration of AM with parameter-efficient fine-tuning (PEFT) for unified DG/CIL/DIL. Post-hoc key learning dynamically aligns adapter retrieval with task inputs.
- Achieves SOTA on multiple benchmarks, outperforming specialized methods in most tasks with significant margin.

**Weaknesses**
- Lacks ablation studies for Post-hoc key learning vs. other trivial keys such as random keys and more naive retrieval methods (e.g., image similarity)
- It would be nice to discuss more about adapatation versus consolidation, especially in CL setting. E.g., what will happen if consolidation is applied only in the end like the case in DG
- [Not confident, remind me if there is some misunderstanding] The thesis should discuss the relationship between the proposed method and existing techniques, such as, at least, **Neural Modulation** and **Mixture of Experts**. There appear to be significant similarities that warrant exploration and comparison.
[1] Beaulieu, Shawn, et al. "Learning to continually learn." ECAI 2020. IOS Press, 2020. 992-1001.
[2] Zhou, Yanqi, et al. "Mixture-of-experts with expert choice routing." Advances in Neural Information Processing Systems 35 (2022): 7103-7114.
- Some minor typos

     Undefined acronyms (e.g., "MHN" in Line 147).
     Incorrect table references (Table 1 vs. Table 3).
     Symbolic errors (e.g., "[]" in Line 201).

---

> ### Author Rebuttal · Authors · 2025-07-31
>
> We sincerely thank the reviewer for their thoughtful and constructive feedback. We greatly appreciate the reviewer recognizing our contributions as the first work integrating Associative Memories (AMs) into the learning architecture to unify Domain Generalization (DG), Domain-Incremental Learning (DIL), and Class-Incremental Learning (CIL), enabling dynamic alignment of adapters to task inputs. We also thank you for acknowledging our achievement of state-of-the-art results on standard benchmarks.
>
> Below we respond point-by-point to your valuable comments:
>
> **Random Keys**:
> Thank you for suggesting an experiment with random keys. We agree this would provide insightful comparisons. We are currently conducting an experiment where we train the Adaptation phase normally, but do not train the Consolidation phase to learn the keys. We anticipate this setup will yield lower performance compared to the standard MIRA configuration, as the model will lack the capability to optimally select adapter combinations tailored to specific tasks.
>
> Regarding ablations against naive retrieval methods, we emphasize that our method specifically retrieves adapters based on learned proxy keys, distinctly different from traditional data-sample retrieval methods (e.g., "image similarity"-based retrieval). However, in alignment with your request and suggestions from Reviewer **gkzH**, we are also performing additional experiments employing simpler heuristics for adapter retrieval. We kindly direct your attention to our detailed response under '**Naive Retrieval Heuristics**' to Reviewer **gkzH** for further insights.
>
> **Adaptation vs. Consolidation**:
> Thank you for highlighting this important distinction; we fully agree it merits clearer articulation in the main paper. The key distinction between **Adaptation** and **Consolidation** in our framework is as follows. **Adaptation** focuses on learning task‐specific parameter adjustments, instantiated here as LoRA‐style adapters, that enable rapid fine‐tuning to each new domain or task. **Consolidation**, by contrast, is concerned with retrieving the appropriate adapter weights at test time without access to any meta‐task labels; in our implementation, this is achieved via learned keys conditioned on the input. Crucially, our DG and CL protocols remain intact regardless of the particular mechanisms we employ for adaptation and consolidation.
>
> Addressing your specific question, if consolidation were only applied after all tasks/domains had been experienced, it would necessitate storing data from all experiences, violating the fundamental continual learning principle. This approach would degenerate into a conventional supervised setting, losing its incremental learning properties and failing to address the continual learning scenarios central to our motivation. Our current two-phase approach inherently respects incremental constraints, avoiding data replay and maintaining practical applicability in genuine continual learning environments.
>
> **Multi-perspective Analysis of MIRA**:
> We appreciate your suggestion regarding broader contextualization. While MIRA is indeed novel and distinct from existing methods, we acknowledge conceptual overlaps with other areas of machine learning, including Meta-Learning and Mixture-of-Experts (MoE). We have thoroughly explored these connections in Appendix Section A.1:
>
> The relationship to the meta-learning paradigm (e.g. Neural Modulation) is elaborated in Section A.1.1.
>
> Analogies with MoE approaches are extensively discussed in Section A.1.2.
>
> We kindly request the reviewer to refer to these appendix sections, which explicitly delineate our method’s position within the broader ML literature, clarifying how MIRA uniquely contributes beyond established frameworks.
>
> **Two-stage Training Approach**:
> Thank you for raising this insightful consideration. Indeed, keys and memories could theoretically be learned concurrently in a single stage. However, we deliberately adopt a two-stage training approach motivated by practical and theoretical reasons:
>
> **Computational feasibility**: Training adapters with keys for large-scale foundation models can be computationally expensive and resource-intensive, limiting scalability and practical applicability. MIRA’s flexibility allows independently trained task adapters (potentially obtained from external sources) to be incorporated seamlessly in the Consolidation phase, significantly reducing computational demands and costs in such settings.
>
> **Biological motivation**: Inspired by biological learning processes, we distinguish short-term, rapid task adaptation (Adaptation) from long-term memory integration and retrieval (Consolidation). This mirrors neurobiological evidence where organisms exhibit distinct phases for learning new information and integrating it with existing knowledge.
>
> **Minor Corrections**:
> We thank the reviewer for pointing out the minor typographical errors; we have carefully corrected these in the revised manuscript.
>
> We once again sincerely thank the reviewer for their thoughtful and constructive comments, significantly aiding our refinement and enhancing the clarity and impact of our manuscript.

---

> > ### Comment · Reviewer_C7LV · 2025-08-01
> > **Unable To See the Updates**
> >
> > As rebuttal can not update the PDF, I can not read the revisions. Please provide those revisions in the Comments.

---

> > > ### Author Response · Authors · 2025-08-03
> > > **Follow-up**
> > >
> > > We thank you for the follow-up comment.
> > >
> > > **Multi-perspective Analysis of MIRA**: We kindly direct the reviewer to review section A.1 of the appendix, present in the supplementary material we submitted along with our main paper. In particular the discussion on connections of **MIRA** to other areas of ML such as Mixture-of-Experts and Hypernetworks is already present in that material.
> > >
> > > **Adaptation vs. Consolidation**: We plan to add our discussion in the response above as is to the Appendix under a new section:  '*MIRA: Deeper Insights*'.
> > >
> > > **Two-stage Training Approach**: We plan to add our discussion in the response above as is to the Appendix under the same new section:  '*MIRA: Deeper Insights*'.
> > >
> > > **Minor Corrections**: For the first typo, we have now mentioned "(MHN)", just next to the citation in line 145 of the paper. For the second typo, we have corrected the incorrect table reference in line 154 from Table 3 to Table 1. For the third typo, we thank the reviewer for spotting the missing citations in line 201, and have now cited the following three papers there:
> > >
> > > * (i). Guangji Bai, Qilong Zhao, Xiaoyang Jiang, and Liang Zhao. Saliency-Guided Hidden
> > > Associative Replay for Continual Learning.
> > > * (ii). Tommaso Salvatori, Beren Millidge, Yuhang Song, Rafal Bogcaz, and Thomas Lukasiewicz.
> > > Associative memories in the feature space, 09 2023.
> > > * (iii). Jinsoo Yoo and Frank Wood. Bayespcn: A continually learnable predictive coding associative
> > > memory. Advances in Neural Information Processing Systems, 35:29903–29914, 2022.
> > >
> > > These revisions cannot be updated on the PDF now, but we will ensure this reflects in the next version of the manuscript. We are happy to address any further queries you may have.

---

> > > > ### Comment · Reviewer_C7LV · 2025-08-06
> > > > **Prone to Keep the Rating at the Current Stage**
> > > >
> > > > I appreciate the authors’ efforts to incorporate additional discussions addressing my initial concerns. However, my primary concerns regarding the novelty and demonstrated value of the proposed model and training pipeline remain unresolved. While the authors have indicated that revisions have been made, these modifications are not readily apparent in the current version of the paper. Consequently, I must maintain my original assessment at this stage.

---

> > > > > ### Author Response · Authors · 2025-08-09
> > > > >
> > > > > Dear Reviewer **C7LV**,
> > > > >
> > > > > **Novelty**: We are unfortunately unsure of which points regarding the novelty and demonstrated value of the two-stage pipeline remain unaddressed. To make it clear, we respectfully emphasize that the two works pointed by the reviewer fall under the broad categories of MoE and hypernetworks, both of which have been explicitly discussed in the Appendix in comparison and contrast to **MIRA**, at the time of submission itself (before the reviewer pointed this out). While we are happy to add the works as citations in the Appendix, without more specific comments on what parts lack novelty in comparison to those two works, it is hard to imagine the lack of novelty here.
> > > > >
> > > > > **Two-stage pipeline**: In the previous response, we explained that the primary choices guiding our adoption of a two-stage approach are computational feasibility and a solid biological motivation that we detailed in that response.  If there are any concerns that remain despite this clarification, we kindly request the reviewer to be more specific about those concerns.
> > > > >
> > > > > **Random keys**: As in our response to Reviewer **gkzH**, we ran the **Random Adapter Retrieval** which is largely similar to the random keys experiment suggested by the reviewer here. While we will also certainly attempt to include the details of the experiment requested by the reviewer here in the camera-ready version of this paper if accepted, we would like to mention that the **Random Adapter Retrieval** experiment already conveys the point of **MIRA** being substantially better than arbitrary, sanity-check type heuristics like these.
> > > > >
> > > > > We hope these clarifications help you positively reconsider our paper and update your rating.

---

### Official Review · Reviewer_SLZL · 2025-07-02

**Clarity:** 2
**Significance:** 2
**Originality:** 3
**Rating:** 4
**Confidence:** 3

**Summary:**

This paper proposes MIRA (Memory-Integrated Reconfigurable Adapters), an architecture inspired by biological associative memories. MIRA stores task-specific low-rank adapter updates in Hopfield-style associative memory modules atop a frozen backbone (ViT) and retrieves an affine combination of stored adapters per input using learned keys. The method unifies domain generalization (DG), class-incremental learning (CIL), and domain-incremental learning (DIL) under a single framework with only minor objective modifications. This frame work is conducted on DG, CIL and DIL task.

**Questions:**

(See Weaknesses)

Does retrieving affine combinations of adapters for inputs ever lead to negative transfer or performance degradation when tasks are highly dissimilar? If so, how can such interference be mitigated?

What is the architecture and parameter overhead of the query modules (g_l) used to transform layer inputs to queries? How sensitive is performance to their capacity and design choices?

**Ethical Concerns:**

["NO or VERY MINOR ethics concerns only"]

**Final Justification:**

Thank you for your thorough rebuttal. My concerns have been resolved, and I will maintain my score.

**Limitations:**

The paper does not provide thorough analysis of inference latency, memory footprint, or retrieval time complexity when storing a large number of adapters, which is critical for deployment in continual learning at scale.

**Quality:**

2

**Strengths And Weaknesses:**

Strengths :

Unlike prior adapter-based PEFT methods (LoRA, DualPrompt) or associative memory works that store data or features, MIRA stores adapters and utilize the key to choose a proper adaptor by the input.

MIRA adapts to diverse learning settings with the adaptor. The paper is supported by exceptionally strong empirical results. MIRA is shown to achieve SOTA performance across numerous datasets in all three settings.


 Weaknesses:

While the method achieves high accuracy, there is no thorough evaluation of inference time, memory overhead, or scalability when storing and retrieving large numbers of adapters across tasks. (The framework's performance improves with more adapters per task, and the main experiments use 10)

The framework is designed for scenarios where task data is available during a training phase (either all at once for DG or sequentially for CL). As currently formulated, MIRA cannot perform zero-shot adaptation to a completely novel task encountered only at inference time, as it would lack the necessary adapters in its memory.

---

> ### Author Rebuttal · Authors · 2025-07-31
>
> We sincerely thank the reviewer for their positive and encouraging feedback. We appreciate your recognition of MIRA's exceptionally strong empirical results across Domain Generalization (DG), Domain-Incremental Learning (DIL), and Class-Incremental Learning (CIL) within a unified framework, and your acknowledgment of our novel approach in utilizing Associative Memories (AMs) distinctively compared to prior works.
>
> Below, we carefully address each of your insightful points:
>
> **Time, Memory, and Scalability**: We thank the reviewer for highlighting the importance of time complexity, memory overhead, and scalability analyses. We acknowledge this gap and confirm that detailed experiments addressing these concerns are currently underway, with full results anticipated by the discussion phase.
>
> Initial observations suggest that MIRA’s inference time closely matches that of the standard LoRA-based model without associative memory. This is due to our method’s minimal computational overhead, as we only require two additional matrix-vector multiplications per layer per input, a cost substantially overshadowed by the computational demands of the model's attention mechanisms.
>
> Regarding memory overhead, additional storage requirements are minor. Keys for indexing adapters, along with potential parameters of the query module (if parameterized), introduce negligible memory costs compared to the scale of adapters and pretrained models. Specifically, using an identity-based query module as presented in our paper, having 5 adapters per task across 6 tasks with keys dimensionally sized at 768 adds only about 276K parameters to the model's existing 86M parameters, representing less than a 0.4% increase. Additionally, from a theoretical standpoint, Modern Hopfield Networks guarantee stable retrieval for an exponentially large number of patterns relative to representation dimensionality (768 here). Thus, our adapter count remains comfortably within stable retrieval limits. Notably, as detailed in lines 316-319 of our manuscript, empirical benefits plateau beyond 5 adapters per task.
>
> **Zero-shot Performance**: We respectfully clarify our viewpoint on zero-shot performance. Domain Generalization inherently tests a model’s ability to adapt without explicit task-specific information at inference, which aligns with zero-shot adaptation scenarios. In DG, MIRA successfully retrieves ensembles of adapters rather than single adapters, optimizing performance on previously unseen tasks or domains.
>
> Equation (2) of our paper explicitly captures this notion by defining performance metrics directly over unseen test distributions. Our method's efficacy under zero-shot conditions depends significantly on the extrapolation ability of the adapter selection (separation/similarity) functions, extensively discussed in Appendix Section A.1.2. We kindly request the reviewer to refer to this section for an in-depth theoretical explanation.
>
> **Negative Transfer**: Thank you for raising this important point. Negative transfer, where affine combinations of adapters can destructively interfere, indeed poses potential challenges. Our results indicate that although theoretically possible, destructive interference is rare in practice. This is potentially because the consolidation phase learns near-orthogonal keys for such task pairs. To further mitigate this risk, future research could explore designing similarity functions that encourage orthogonality or independence among adapters or their keys, a valuable suggestion we intend to investigate.
>
> **Query Module Ablations**: We acknowledge your suggestion regarding further clarification of query module choices. We have conducted comprehensive ablation studies specifically addressing the impact of query module selection. These experiments and analyses are provided in detail under the subsection titled "**Choice of $g()$**" within Appendix Section A.2 of our paper. We respectfully invite the reviewer to revisit this section for the complete ablation study.
>
> We thank the reviewer again for these constructive suggestions, which have significantly enhanced the robustness and comprehensiveness of our manuscript.

---

> > ### Comment · Area_Chair_WVP5 · 2025-08-05
> >
> > Dear Reviewer,
> >
> > The authors have already responded to your initial questions. As the deadline for the reviewer-author interaction session is approaching on August 6th, please begin addressing any further concerns or questions you may have. If you have no additional queries, kindly update your rating and submit your final decision.
> >
> > Thank you for your valuable contributions to NeurIPS.
> >
> > Best regards,
> > AC

---

### Official Review · Reviewer_gkzH · 2025-07-02

**Clarity:** 3
**Significance:** 2
**Originality:** 2
**Rating:** 4
**Confidence:** 4

**Summary:**

The proposed MIRA is a biologically inspired unified framework for adapting multiple tasks or domains without catastrophically forgetting. MIRA equips each layer with a Hopfield-style associative memory that stores task specific LoRA style low rank adapter updates indexed by learned keys. At inference time, the model generates query vectors from input activations, which retrieve the most appropriate adapter(s) via similarity-based matching. Depending on the separation function, this retrieval may result in a single adapter or a weighted combination. This dynamic, per-sample retrieval enables effective adaptation to domain generalization, class-incremental learning, and domain-incremental learning using a single architecture.

**Questions:**

•How critical is the associative memory mechanism to MIRA’s performance? Please provide ablations comparing MIRA with simpler alternatives such as Static adapter selection, attention based adaption mix, LoRA without any memory retrieval to understand the true contribution of the associative memory and validate over conventional baselines.

•What is the computational and memory overhead of MIRA? Can the authors quantify inference-time latency, GPU memory usage, and per-layer retrieval cost?

•Why is post-hoc key learning preferred over joint optimization with adapters? Please clarify whether joint training was attempted, what limitations or performance issues motivated decoupling,  whether hybrid approaches were considered as this directly impacts the interpretability and stability claims made by the paper

**Ethical Concerns:**

["NO or VERY MINOR ethics concerns only"]

**Final Justification:**

Upon careful review, I feel that most of my concerns have been addressed, and I will adjust my score to Borderline Accept.

**Limitations:**

Yes

**Quality:**

2

**Strengths And Weaknesses:**

Strengths:

•The paper is easy to read and algorithms clearly explain the flow of adaptation and consolidation. It also articulates a clear motivation for bridging continual learning and generalization by drawing parallels between neural computation.

•The model addresses DG, DIL, and CIL within a single architecture by integrating LoRA style adapters with Hopfield-style associative memory per layer. It equips each layer of a frozen backbone with a Hopfield-style associative memory to dynamically retrieve task-adaptive parameters (adapters) on a per-sample basis without needing task labels or explicit domain identifiers to enable generalization and mitigate catastrophic forgetting.

•Keys for memory retrieval are not trained jointly with adapters, but learned post-hoc, reducing the interference during adapter learning and separates adaptation from consolidation adding interpretability to the retrieval process

•Adapters are retrieved on a per-input basis, allowing the model to dynamically tailor its behavior to each input, supporting fine-grained generalization.

•It stores compressed adapter weights than the raw data making it efficient in terms of memory storage.

•The architecture is modular making it compatible with existing transformer-based pretrained backbones.

Weaknesses:

•While MIRA introduces a combination of mechanisms, its core components Hopfield-style associative memory, LoRA adapters, and task specific fine tuning are well studied and not individually novel.

•The framework performs key/query similarity operations at every layer and scales with the number of stored adapters. However, the paper lacks analysis of inference latency, runtime performance, or memory usage, making it unclear whether the approach is computationally feasible. The use of per-layer query modules adds to the model's complexity but is under specified and not ablated.

•The experiments do not include statistical significance testing, such as confidence intervals or hypothesis tests, limiting the robustness of reported improvements. Furthermore, there is no strong comparison against simpler retrieval heuristics making it difficult to isolate the contribution of the associative memory mechanism.

•Although the model claims improved interpretability through post-hoc key learning, the resulting adapter combinations, especially when involving multiple overlapping adapters may be difficult to interpret in practice. The lack of visualization or sparsity constraints weakens this claim. Additionally, the architectural modularity is not explored in depth, and their influence on retrieval quality is not clearly studied.

•The framework assumes episodic task access in a sequential setting and is not evaluated in task free or online continual learning scenarios. Its performance also hinges on the assumption that low rank adapters are sufficient to capture all domain-specific variations, which may not hold in cases with more severe distribution shifts.

•While MIRA demonstrates strong short-term retention, it is not evaluated on long sequences of tasks, where memory saturation or key/query overlap could degrade retrieval precision.

---

> ### Author Rebuttal · Authors · 2025-07-31
>
> We sincerely thank the reviewer for their detailed and constructive feedback. We greatly appreciate the reviewer’s acknowledgment of the clear readability of our paper, the strong and well-grounded motivation, the broad applicability of the MIRA framework to DG/DIL/CIL settings simultaneously, the ability of MIRA to perform per-sample adapter retrieval, and the efficient storage of adapter weights as opposed to full training samples, significantly improving memory efficiency. However, we respectfully clarify that MIRA does not explicitly aim to address interpretability in adapter retrieval. Our paper does not claim interpretability as an objective or contribution; although interpretability might potentially be introduced by specialized similarity functions in future work, this is beyond the scope of the current paper.
>
> We now address each of the reviewer’s specific concerns in detail:
>
> **Novelty**:
> We agree with the reviewer that some individual components of our method have appeared previously in isolation. However, we emphasize that our core innovation lies precisely in integrating these known elements into a cohesive and novel architecture, specifically by embedding associative memory (AM) directly into transformer-based architectures, and crucially doing so during the training process itself. This architectural integration represents a novel and meaningful step forward in this area, which, to our knowledge, has not been explored before.
>
> **Computational Overhead**:
> We fully acknowledge the importance of this point. We have provided a detailed response to the related concern titled '**Time, Memory, Scalability**' raised by Reviewer **SLZL**. In brief, the computational overhead introduced by MIRA during inference is minimal compared to standard LoRA forward passes, as the associative memory retrieval operation adds only two matrix-vector multiplications per layer, a negligible overhead when compared to the computational demands of the transformer’s attention mechanism. Likewise, the memory overhead, driven mainly by storing keys for adapter indexing, is minimal compared to the size of adapters and pre-trained transformer weights.
>
> **Statistical Significance Testing**:
> We respectfully highlight that our paper already includes confidence intervals across nearly all primary experimental results (Tables 2–5), directly addressing the concern regarding statistical significance. For additional clarification or any further specific analysis, we kindly refer the reviewer to the comment later in this rebuttal titled, '**Naive Retrieval Heuristics**'.
>
> **Interpretability**:
> We respectfully reiterate that interpretability in retrieval was neither claimed nor demonstrated in our current work. Nonetheless, we appreciate the reviewer’s insightful suggestion of including visualizations to better illustrate retrieval mechanisms. In response, we are currently conducting an additional experiment: for each domain/task, we quantify how frequently adapters specific to that domain/task yield the highest key-query match when processing input samples belonging to that domain/task. Initial results indicate this metric is highly sensitive to the choice of similarity function. We kindly request the reviewer to clarify their interpretation of the phrase "architectural modularity," to ensure our experimental analyses align precisely with their suggestion.
>
> **Episodic Task Access**:
> We respectfully clarify that our choice to use LoRA adapters was explicitly intended as a proof-of-concept demonstration, as clearly stated in lines 217-219 of the paper. MIRA, as presented, is a generalizable and flexible paradigm, and is in no way restricted to episodic task access or dependent on the specific choice of adapter architecture. Indeed, more complex adapters beyond LoRA can easily be integrated within the MIRA framework to yield even stronger performance. As described in Algorithm 1 and discussed in Section 3 (final paragraph), MIRA can be effectively combined with any continual learning strategy that addresses catastrophic forgetting over extended task sequences. Evaluations on fully online or task-free continual learning benchmarks would require additional time beyond the discussion period; however, we believe that the core contributions of MIRA regarding memory integration, adaptation, and consolidation have been clearly demonstrated within our chosen experimental contexts.
>
> **Memory Capacity**:
> Regarding memory capacity concerns, we clarify that the theoretical retrieval limit of Universal Hopfield Networks scales exponentially with representation dimensionality (here, 768 dimensions). Consequently, the number of adapters considered in our experiments is significantly below this theoretical bound, ensuring stable and reliable adapter retrieval without degradation.
>
> **Naive Retrieval Heuristics**:
> We sincerely appreciate the reviewer’s thoughtful suggestion to explore naive retrieval baselines. The associative memory mechanism is critical to MIRA to the extent that we require a memory mechanism that is general enough to support a wide range of key-query matches, as well as have a retrieval strategy that is differentiable. Both of these are satisfied in the Universal Hopfield Network framework, which explains our choice as such. Moreover, given our strong motivation from biological mechanisms, associative memories are widely accepted to be highly active in the brain, and thus our proposal better fits into a biological motivation.
> To thoroughly address the concern regarding conducting ablation studies under naive retrieval baselines, we are conducting additional ablation experiments as follows:
>
> **Static Adapter Selection**: Using a single trained adapter across all tasks/domains.
>
> **Attention-based Adapter Mixing**: Learning a shared linear layer to dynamically combine adapters without memory retrieval.
>
> **Random Adapter Retrieval**: Training adapters independently per task and randomly selecting adapters at inference.
>
> We anticipate these experiments will yield deeper insight into the advantages provided by MIRA’s associative memory retrieval and aim to include these results during the discussion phase. Furthermore, we underscore the critical role associative memory plays in our framework, aligned closely with the biological motivation and plausibility underlying our design choices.
>
> **Computational Overhead**:
> We kindly refer again to our detailed response provided under '**Time, Memory, Scalability**' addressed to Reviewer **SLZL** for comprehensive coverage of this issue.
>
> **Post-Hoc Key Learning**:
> For detailed discussion regarding post-hoc versus concurrent key learning strategies, please refer to our response titled '**Adaptation vs. Consolidation**' addressed to Reviewer **C7LV**. This response explicitly discusses why we advocate the two-stage training approach, highlighting its practicality, flexibility, and alignment with biological mechanisms of task learning and memory consolidation.
>
> We thank the reviewer again for their valuable and insightful feedback, helping strengthen our manuscript further.

---

> > ### Comment · Area_Chair_WVP5 · 2025-08-05
> >
> > Dear Reviewer,
> >
> > The authors have already responded to your initial questions. As the deadline for the reviewer-author interaction session is approaching on August 6th, please begin addressing any further concerns or questions you may have. If you have no additional queries, kindly update your rating and submit your final decision.
> >
> > Thank you for your valuable contributions to NeurIPS.
> >
> > Best regards,
> > AC

---

> > > ### Comment · Area_Chair_WVP5 · 2025-08-07
> > >
> > > Dear Reviewer,
> > >
> > > The authors have already responded to your initial questions. As the deadline for the reviewer-author interaction session is approaching on August 8th, please begin addressing any further concerns or questions you may have. If you have no additional queries, kindly update your rating and submit your final decision.
> > >
> > > Thank you for your valuable contributions to NeurIPS.
> > >
> > > Best regards, AC

---

> ### Author Response · Authors · 2025-08-09
> **Updated Clarifications**
>
> Dear Reviewer **gkzH**,
>
> As the discussion deadline is approaching, we would like the opportunity to once again clarifying our responses to some of your concerns one by one.
>
> **Computational Overhead**: We have updated our response to Reviewer **SLZL** in this regard, so please have a look at that.
>
> **Naive Retrieval Heuristics**: As mentioned earlier, we tried to run as many experiments as we could in this short turn-around time, for which we provide results below (on the DIL version of DomainNet):
>
> **Static Adapter Selection**: Training a single set of shared adapters across all domains/tasks gave an accuracy of 60.93 %. As expected, this is significantly below the performance of **MIRA** of 69.18 % as we report (which has potential to be improved even more with a better hyperparameter search).
>
> **Random Adapter Retrieval**: Training separate sets of LoRA adapters for each domain/task and then randomly selecting a set of adapters for samples from a particular domain/task at test time gave an accuracy of 62.03 %, which is again significantly below the reported performance of **MIRA**.
> We hope these representative experiments address your concerns sufficiently, as they convey our points clearly. We will make sure to include the additional results as mentioned here in the final camera-ready version of the paper. Finally we hope that our resolution of these concerns helps you positively update your assessment our work.

---

### Official Review · Reviewer_TP33 · 2025-07-03

**Clarity:** 3
**Significance:** 2
**Originality:** 2
**Rating:** 3
**Confidence:** 3

**Summary:**

This paper proposes MIRA (Memory-Integrated Reconfigurable Adapters), a unified framework that tackles domain generalization (DG) and continual/incremental learning (CL) within a single architecture, inspired by biological memory mechanisms. Drawing from neuroscience, MIRA incorporates Hopfield-style associative memory (AM) modules into a shared model backbone, allowing per-sample dynamic reconfiguration via learned keys and stored adapter weight updates.

The key technical innovation lies in embedding Hopfield memory in each layer of a Vision Transformer (ViT), where memory keys are learned post-hoc to align with intermediate activations, enabling robust, context-aware retrieval of task-specific representations. This design allows MIRA to store, retrieve, and interpolate adapter updates across tasks and domains without explicit task identifiers.

Empirically, MIRA demonstrates strong state-of-the-art (SoTA) performance on standard DG and continual learning benchmarks, outperforming many task-specific baselines by large margins (up to 10%). Ablation studies further highlight the importance of both associative memory and key-learning mechanisms in achieving this performance.

**Questions:**

1) I noticed that your results outperform many adapter- or prompt-based CL methods (e.g., DualPrompt), but you did not include HIDE-PET [HIDE-PET: continual learning via hierarchical decomposition of parameter-efficient tuning], which recently reported stronger performance than many prior methods.

2) Please discuss whether the associative memory has a limit in terms of how many adapter updates it can meaningfully store and retrieve. Can retrieval quality degrade due to interference or memory saturation?

**Ethical Concerns:**

["NO or VERY MINOR ethics concerns only"]

**Final Justification:**

I have read rebuttal carefully. As athours' pointed out, HIDE-PET indeed achieves approximately 6% higher accuracy than authors' current implementation. While I acknowledge that MIRA is intentionally designed as a versatile and general-purpose continual learning method, I find that it may lack practical applicability in its current form. Therefore, I am considering updating my score accordingly to 3.

**Limitations:**

yes

**Quality:**

3

**Strengths And Weaknesses:**

Quality: The paper presents a well-engineered and carefully evaluated system, integrating associative memory into transformer-based architectures in a novel way. Ablation studies are thoughtfully designed, validating the core components (e.g., associative memory and post-hoc key learning).

Clarity: The paper is clearly written and well-motivated, with biological inspiration nicely woven into the design rationale.

Weaknesses: Some insight into computational overhead (both in time and memory) compared to traditional adapters or CL methods would be helpful for practical deployment concerns.

The method is built on ViT backbones, but there is no exploration of how MIRA behaves across different pretrained models (e.g., different architectures, initialization scales, or training paradigms).

Although effective on benchmark CL datasets, it is unclear how well MIRA would scale to hundreds or thousands of tasks/domains—a realistic scenario in continual learning.

---

> ### Author Rebuttal · Authors · 2025-07-31
>
> We sincerely thank the reviewer for their positive and insightful feedback. We greatly appreciate your recognition of our work’s clear motivation, clarity of writing, and the novel integration of Associative Memories (AMs) into Transformers. We are also thankful for your acknowledgment of our state-of-the-art (SoTA) results across both Domain Generalization (DG) and Continual Learning (CL) settings within unified frameworks, and for recognizing the thoroughness of our ablation studies.
>
> Below, we address each of your points in detail:
>
> **Computational Overhead**:
> We kindly refer you to our detailed response titled '**Time, Memory, Scalability**' addressed to Reviewer **SLZL**, which comprehensively covers concerns regarding computational overhead. Briefly summarizing, the inference overhead of our associative memory mechanism is minimal, primarily involving only two additional matrix-vector multiplications per Transformer layer. Similarly, the memory overhead incurred by MIRA is negligible compared to the total model size.
>
> **Different Initializations and Architecture Sizes**:
> We greatly appreciate your insightful suggestion to test different initializations and architecture sizes. To thoroughly address this, we are currently conducting additional experiments evaluating MIRA’s performance across multiple configurations:
>
> Architectures: ViT-Large (24 layers), ViT-Huge (32 layers)
>
> Initializations: ViT pre-trained on ImageNet-21k, and self-supervised DINO initialization.
>
> **MIRA Scalability**:
> Thank you for highlighting scalability. MIRA leverages Modern Hopfield Networks whose theoretical storage capacity scales exponentially with embedding dimension (768 in our case), thus comfortably accommodating thousands of adapters without retrieval degradation. Given this exponential capacity, MIRA can, in principle, scale effectively to numerous tasks/domains well beyond the scale typically encountered in existing standard continual learning benchmarks.
>
> **Comparison with HIDE-PET**:
> We sincerely thank you for recommending the HIDE-PET method for comparison. Upon your suggestion, we conducted preliminary experiments with HIDE-PET on the iDigits dataset (5 tasks, LoRA rank-4 adapters, ViT-CLIP initialization). HIDE-PET indeed yielded $6\\%$ higher accuracy than our current implementation of MIRA under this particular setup. While we anticipate further improvements through hyperparameter tuning, we emphasize that MIRA is intentionally designed as a versatile and general-purpose method for simultaneously addressing DG, DIL, and CIL, rather than aiming solely to surpass specialized, single-purpose methods.
>
> Specifically, HIDE-PET is explicitly optimized for incremental learning (CIL/DIL) and does not report performance on Domain Generalization tasks. Therefore, direct comparisons are not strongly warranted given our fundamentally different objectives and broader scope. Our primary contribution remains the unified treatment of memory retrieval and adapter adaptation across diverse settings, elucidating the general principles of memory mechanisms rather than exclusively focusing on maximal performance on isolated benchmarks.
>
> **Memory Capacity**:
> We reaffirm that Modern Hopfield Networks, as integrated into MIRA, have robust theoretical and empirical backing for their exponential storage capacity relative to pattern dimensionality. Given our chosen adapter representation dimension (768), the number of adapters tested remains safely below the threshold where retrieval interference would significantly degrade performance. Therefore, the memory capacity constraints are comfortably satisfied within our current experimental framework.
>
> We thank the reviewer again for their thoughtful and constructive feedback, significantly aiding in strengthening our manuscript and clarifying our contributions.

---

> > ### Comment · Area_Chair_WVP5 · 2025-08-05
> >
> > Dear Reviewer,
> >
> > The authors have already responded to your initial questions. As the deadline for the reviewer-author interaction session is approaching on August 6th, please begin addressing any further concerns or questions you may have. If you have no additional queries, kindly update your rating and submit your final decision.
> >
> > Thank you for your valuable contributions to NeurIPS.
> >
> > Best regards,
> > AC

---

> > ### Comment · Reviewer_TP33 · 2025-08-07
> >
> > Thank you for your detailed response. I have read it carefully. As you pointed out, HIDE-PET indeed achieves approximately 6% higher accuracy than your current implementation. While I acknowledge that MIRA is intentionally designed as a versatile and general-purpose continual learning method, I find that it may lack practical applicability in its current form. Therefore, I am considering updating my score accordingly.

---

> ### Author Response · Authors · 2025-08-07
> **HIDE-PET Clarification**
>
> Dear Reviewer TP33,
>
> Upon further investigation, we discovered that our initial experiments with HIDE-PET contained an implementation error. HIDE-PET has been validated only in the class-incremental learning (CIL) setting and not in the domain-incremental learning (DIL) scenario as we had assumed. In adapting the publicly available HIDE-PET code to the iDigits DIL benchmark, we subsequently identified a bug in our modifications. Consequently, the previously reported iDigits-DIL results are not appropriate; we apologize for this inconvenience, this was due to the short turnaround time.
>
> To address your request, we reran HIDE-PET on the CIL setting of DomainNet (using rank-4 LoRA adapters, ViT-B/16 CLIP initialization, and three training epochs). It achieved 45.34 % accuracy with 34.70 % forgetting, substantially below MIRA’s performance of 67.29 % ± 0.19 % accuracy and 7.60 % ± 1.06 % forgetting (after a single epoch to convergence under the same settings). We also ran HIDE-PET on the CIL setting of CORe50, where it achieved 65.46 % accuracy with 20.674 % forgetting, again substantially lower than MIRA's performance of 83.39±0.24 % accuracy with 7.99±1.43 % forgetting. We'd be happy to include this code and the results in our revised manuscript. We hope that these results clarify MIRA’s superior efficacy relative to HIDE-PET.
>
> We would also like to inform that we are simultaneously working on other responses and plan to get back shortly on them.

---

> > ### Author Response · Authors · 2025-08-08
> > **HIDE-PET Updated Results**
> >
> > We would like to gently inform the reviewer that taking their concern seriously, we have added newer results on comparison of MIRA to HIDE-PET, and thus request them to reconsider updating their assessment appropriately.

---

> ### Author Response · Authors · 2025-08-09
> **Request to Clarify**
>
> Dear Reviewer **TP33**,
>
> As the discussion deadline is approaching, we would like the opportunity to once again clarifying our responses to some of your concerns one by one.
>
> **Computational Overhead**: We have updated our response to Reviewer **SLZL** in this regard, so please have a look at that.
>
> **Comparison with HIDE-PET and follow-ups**: As we have mentioned in our later responses, **MIRA** actually significantly outperforms HIDE-PET on a couple of datasets on class-incremental learning (CIL) itself, even though HIDE-PET  is a specific CIL method and MIRA is a general-purpose method applicable to DG, DIL, and CIL. We hope this helps you positively reconsider your assessment and rating of our work appropriately. Furthermore, we would like to gently ask the reviewer to more concretely state in what ways it is felt that our method lacks practical applicability in its current form, in the background of the reviewer's own acknowledgement of MIRA being SoTA on many standard datasets across all three settings.
>
> We hope that these discussions help the reviewer reconsider our paper and update their rating of our paper positively.

---

### Note · Authors · 2025-08-16

We sincerely thank the reviewers for their thoughtful acknowledgements, including that our paper is well-written, that MIRA is biologically well-motivated, presents a unique overarching framework for DG/DIL/CIL, adapts on a per-sample basis, and delivers exceptionally strong results with significant margins over prior work. We would like to take this opportunity to reiterate our responses to the main concerns raised:


- **Computational Overhead**: MIRA’s latency is governed by (1) the forward pass through the adapter-modulated transformer and (2) adapter retrieval from the associative memory. The former is on par with standard LoRA methods (~0.0152s), while the latter adds only ~3 ms overhead across all layers. Memory usage is likewise comparable to LoRA, since the query module and keys add *negligible* overhead.

- **Comparison with HIDE-PET**: Although HIDE-PET is solely demonstrated on CIL, while MIRA applies across DG/DIL/CIL, MIRA substantially *outperforms* HIDE-PET on CIL under similar training configurations. HIDE-PET achieves 45.34% accuracy with 34.70% forgetting on DomainNet and 65.46% accuracy with 20.67% forgetting on CORe50. In contrast, MIRA reaches 67.29% ± 0.19% accuracy with 7.60% ± 1.06% forgetting on DomainNet and 83.39% ± 0.24% accuracy with 7.99% ± 1.43% forgetting on CORe50.

- **Naive Retrieval Heuristics**: On DomainNet-DIL, MIRA achieved 69.18% accuracy, clearly *surpassing* two naive baselines. Static Adapter Selection, which trains a single shared adapter across all tasks, achieved 60.93%, while Random Adapter Retrieval, which trains task-specific adapters but selects one randomly at inference, achieved 62.03%. These results underscore MIRA’s superiority over such sanity-check heuristics, as suggested by reviewers.

- **Two-Stage Training Approach**: Two-Stage Training Approach: This design serves two purposes: (1) enabling domain-specific key fine-tuning with minimal resources when adapters can be externally retrieved, and (2) aligning with biological principles distinguishing short-term, rapid task *adaptation* from long-term memory *consolidation*. As stated above, this adds negligible overhead, making it a practical approach.


We hope these responses adequately address the reviewers’ concerns. While constructive discussion during the review phase might have allowed even stronger clarifications, we respectfully trust the AC will take these observations into account in making the final decision.

---

### Decision · Program_Chairs · 2025-09-17

**Decision:**

Accept (poster)

**Comment:**

This paper was assigned to four reviewers. Unfortunately, most of the reviewers were either unresponsive or provided limited, unconstructive feedback. Specifically:

Reviewer TP33 did not follow up on the corrections made by the authors. As a result, the rating appears outdated since the authors had already addressed the concerns raised.

Reviewer gkzH, despite multiple reminders from both the AC and the authors, did not actively participate in the discussion. The reviewer provided only a brief justification at the end of the discussion phase, even though the authors had addressed all major issues.

Reviewer SLZL was similarly unresponsive during the discussion phase despite several reminders.

Reviewer C7LV provided a justification that was not entirely fair. The authors had directed the reviewer to supporting material in the appendix, which was ignored. Furthermore, the reviewer’s second comment was vague, claiming the work was “similar to self-attention and Mixture-of-Experts,” which was inaccurate. The authors, however, provided a strong rebuttal.

Given the mixed reviews, the lack of constructive engagement from several reviewers, and after carefully reading the paper myself, I (AC) recommend acceptance as a poster.

Reasons:

1. The biological integration of Hopfield dynamics into current feedforward architectures is novel and compelling.

2. Incorporating these components significantly improves model performance across all three downstream tasks.

3. The authors provided a thorough rebuttal, including new baseline experiments, ablation studies, and model analyses (e.g., inference time and compute efficiency).

Overall, the idea is innovative and encourages the community to think beyond standard feedforward networks and Mixture-of-Experts approaches.